# A Geometric Analysis of Multi-label Learning under the Pick-all-label Loss through Neural Collapse

## Abstract

In this study, we explore multi-label learning, an important subfield of supervised learning that aims to predict multiple labels from a single input data point. This research investigates the training of deep neural networks for multi-label learning through the lens of neural collapse, an intriguing phenomenon that occurs during the terminal phase of training. Previously, neural collapse (NC) has been investigated both theoretically and empirically in the context of multi-class classification. For last-layer features, it has been demonstrated that (i) the variability of features within classes collapses to zero, and (ii) the feature means between classes become maximally and equally separated. In this work, we demonstrate that the NC phenomenon can be extended to multi-label learning, revealing that the "pick-all-label" training formulation for multi-label learning exhibits the NC phenomenon in a more general context. Specifically, under the natural analog of the unconstrained feature model, we establish that the only global minimizers of the pick-all-label loss display the same equi-angular tight frame (ETF) geometry. Additionally, scaled average of the ETF are used to represent the features of samples with multiple labels. We also provide empirical evidence to support our investigation into training deep neural networks on multi-label datasets, resulting in improved training efficiency.

## 1 Introduction

In recent years, we have witnessed tremendous success in using deep learning for classification problems. This success can be attributed in part to the deep model's ability to extract salient features from data. While deep learning has also been fruitfully applied to `M-lab`, the structures of the learned features in the `M-lab` regime is less well-understood. The motivation of this work is to fill this gap in the literature by understanding the geometric structures of features from `M-lab`, with the goal of improving the training and generalization in `M-lab`.

Recently, for `M-clf` using overparamterized deep networks, an intriguing phenomenon has been observed in the terminal phase of training, in which the last-layer features and classifiers collapse to simple but elegant mathematical structures: all training inputs are mapped to class-specific points in feature space, and the last-layer classifier converges to the dual of the features' class means while attaining the maximum possible margin with a simplex equiangular tight frame (Simplex ETF) structure (see the top line of Figure 1). This phenomenon, dubbed *Neural Collapse* (NC), persists across a variety of different network architectures, datasets, and even problem formulations Papyan et al. (2020); Han et al. (2022). The NC phenomenon has been widely observed and analyzed theoretically in the context of `M-clf` learning problems from the perspectives of training and optimization Papyan et al. (2020); Zhu et al. (2021), transfer learning Galanti et al. (2022b), and robustness Papyan et al. (2020); Ji et al. (2022), where the line of study has significantly advanced our understanding of representation structures for `M-clf` using deep networks. However, it remains unclear if a generalized version of the NC phenomenon with new geometry emerges in training deep neural networks for `M-lab`. Further study in this area would enhance our understanding of deep learning for `M-lab`.

**Our contributions.** In this work, we demonstrate a general version of the NC phenomenon in `M-lab`, and our study provides new insights into training stage of deep neural networks for the `M-lab` problem. In particular, our contributions can be summarized as follows.

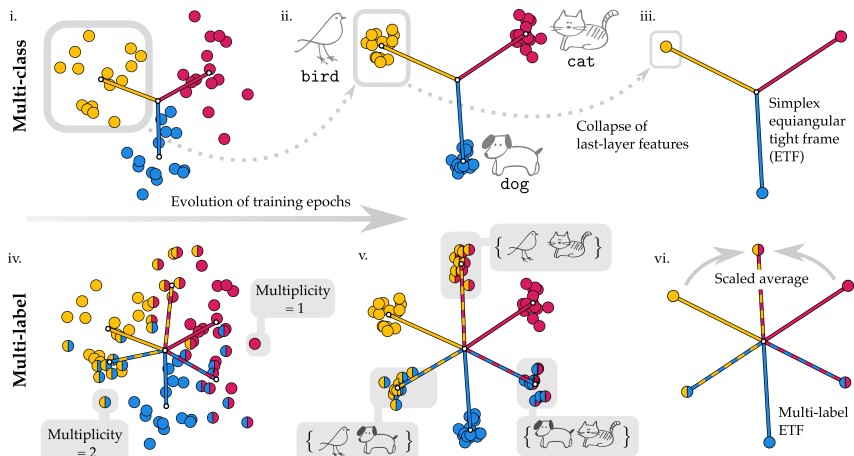

Figure 1: **An illustration of neural collapse for `M-clf` (top row) vs. `M-lab` (bottom row) learning** under the unconstrained feature model. We consider a simple setting with the number of classes $K = 3$. The individual panels are scatterplots showing the top two singular vectors of the last-layer features $\mathbf{H}$ at the beginning (left) and end (right) stages of training. The solid (resp. dashed) line segments represent the centroid of the multiplicity $= 1$ (resp. $= 2$) features with the same labels. *Panel i-iii*. As the training progresses, the last-layer features of samples corresponding to a single label, e.g., `bird`, collapse tightly around its centroid. *Panel iv-vi*. The analogous phenomenon holds in the multi-label setting. *Panel iv*. A training sample has multiplicity $= 1$ (resp. $= 2$) if it is labeled only by a singleton (resp. doubleton) set. *Panel vi*. At the end stage of training, the Multiplicity-2 centroid for {`bird`, `cat`} is a scaled average of the centroids representing {`bird`} and {`cat`} and so on.

- **Multi-label neural collapse phenomenon.** We show that the last-layer features and classifier learned via overparameterized deep networks exhibit a more general version of NC which we term it as *multi-label neural collapse* (`M-lab` NC). In particular, while the features associated with labels of Multiplicity-1 are still forming the Simplex ETF, the *high-order* Multiplicity features are *scaled average* of their associated features in Multiplicity-1. We call the new structure *multi-label ETF*, and we demonstrate its prevalence on training practical neural networks for `M-lab`. Moreover, we show that the multi-label ETF only requires balanced training samples in each class within *the same* multiplicity, and *allows class imbalanced-ness* across different multiplicities.

- **Global optimality and benign landscapes.** Theoretically, we show that the `M-lab` NC phenomenon can be justified based on the unconstrained feature model, where the last-layer features are treated asunconstrained optimization variables Zhu et al. (2021). Under such an assumption, we study the global optimality of a commonly used pick-all-label loss for `M-lab`, showing that all global solutions exhibit the properties of `M-lab` NC. We also prove that the optimization landscape has benign strict saddle properties so that global solutions can be efficiently achieved.

**Related work on multi-label learning.** In contrast to `M-clf`, where each sample has a single label, in `M-lab` the samples are tagged with multiple labels. This presents theoretical and practical challenges unique to the multi-label regime. From the practical side, many modern deep neural network architectures have been successfully adapted to the multi-label task Chang et al. (2020); Lanchantin et al. (2021); Ridnik et al. (2023). However, the methods often suffer from the challenges of imbalanced training data, given that high Multiplicity labels are scarce. On the theory side, consistency of surrogate methods for `M-lab` has been initiated by Gao & Zhou (2011) and followed up by several works in Menon et al. (2019); Dembczynski et al. (2012); Zhang et al. (2020a); Blondel et al. (2020). Many other concepts from classical learning theory have also been extended successfully to the `M-lab` regime, e.g., Vapnik-Chervonenkis theory and sample-compression schemes Samei et al. (2014a;b), (local) Rademacher complexity Xu et al. (2016); Reeve & Kaban (2020), and Bayes-optimal prediction Cheng et al. (2010). However, to the best of our knowledge, no work has previously analyzed the geometric structure arisen in multi-label deep learning. Our work closes this gap, providing a generalization of the neural collapse phenomenon to multi-label learning.

**Related work on neural collapse.** The phenomenon known as NC was initially identified in recent groundbreaking research Papyan et al. (2020); Han et al. (2022) conducted on `M-clf`. These studies provided empirical evidence demonstrating the prevalence of NC across various network architectures and datasets. The significance of NC lies in its elegant mathematical characterization of learned representations or features in deep learning models for `M-clf`. Notably, this characterization

is independent of network architectures, dataset properties, and optimization algorithms, as also highlighted in a recent review paper Kothapalli (2023). Subsequent investigations, building upon the "unconstrained feature model" Mixon et al. (2022) or the "layer-peeled model" Fang et al. (2021), have contributed theoretical evidence supporting the existence of NC. This evidence pertains to the utilization of a range of loss functions, including cross-entropy (CE) loss Lu & Steinerberger (2022); Zhu et al. (2021); Fang et al. (2021); Yaras et al. (2022), mean-square-error (MSE) loss Mixon et al. (2022); Zhou et al. (2022a); Tirer & Bruna (2022); Rangamani & Banburski-Fahey (2022); Wang et al. (2022); Dang et al. (2023), and CE variants Graf et al. (2021); Zhou et al. (2022b). More recent studies have explored other theoretical aspects of NC, such as its relationship with generalization Hui et al. (2022); **?**); Galanti et al. (2022a); Galanti (2022); Chen et al. (2022), its applicability to large classes Liu et al. (2023); Gao et al. (2023), and the progressive collapse of feature variability across intermediate network layers Hui et al. (2022); Papyan (2020); He & Su (2023); Rangamani et al. (2023). Theoretical findings related to NC have also inspired the development of new techniques to improve practical performance in various scenarios, including the design of loss functions and architectures Yu et al. (2020); Zhu et al. (2021); Chan et al. (2022), transfer learning Li et al. (2022); Xie et al. (2022), imbalanced learning Fang et al. (2021); Xie et al. (2023); Yang et al. (2022); Thrampoulidis et al. (2022); Behnia et al. (2023); Zhong et al. (2023); Sharma et al. (2023), and continual learning Yu et al. (2023); Yang et al. (2023).

**Basic notations.** Throughout the paper, we use bold lowercase and upper letters, such as $\boldsymbol{a}$ and $\boldsymbol{A}$, to denote vectors and matrices, respectively. Non-bold letters are reserved for scalars. For any matrix $\boldsymbol{A} \in \mathbb{R}^{n_1 \times n_2}$, we write $\boldsymbol{A} = [\boldsymbol{a}_1 \quad \dots \quad \boldsymbol{a}_{n_2}]$, so that $\boldsymbol{a}_i$ ($i \in \{1, \dots, n_2\}$) denotes the $i$-th column of $\boldsymbol{A}$. Analogously, we use the superscript notation to denote rows, i.e., $(\boldsymbol{a}^j)^\top$ is the $j$-th row of $\mathbf{A}$ for each $j \in \{1, \dots, n_1\}$ with $\mathbf{A}^\top = [\boldsymbol{a}^1 \quad \dots \quad \boldsymbol{a}^{n_1}]$. For an integer $K > 0$, we use $\boldsymbol{I}_K$ to denote a identity matrix of size $K \times K$, and we use $\mathbf{1}_K$ to denote an all-ones vector of length $K$.

## 2 PROBLEM FORMULATION

We start by reviewing the basic setup for training deep neural networks, and later specialize to the problem of M-lab with $K$ number of classes. Given a labelled training instance $(\boldsymbol{x}, \boldsymbol{y})$, the goal is to learn the network parameter $\boldsymbol{\Theta}$ to fit the input $\boldsymbol{x}$ to the corresponding training label $\boldsymbol{y}$ such that

$$\boldsymbol{y} \approx \psi_{\boldsymbol{\Theta}}(\boldsymbol{x}) = \underbrace{\boldsymbol{W}_L}_{\text{linear classifier } \boldsymbol{W}} \cdot \ \sigma\underbrace{(\boldsymbol{W}_{L-1} \cdots \sigma(\boldsymbol{W}_1 + \boldsymbol{b}_1) + \boldsymbol{b}_{L-1})}_{\text{feature } \boldsymbol{h} = \phi_{\boldsymbol{\theta}}(\boldsymbol{x})} + \boldsymbol{b}_L, \tag{1}$$

where $\boldsymbol{W} = \boldsymbol{W}_L$ represents the last-layer linear classifier and $\boldsymbol{h}(\boldsymbol{x}) = \phi_{\boldsymbol{\theta}}(\boldsymbol{x}_{k,i})$ is a deep hierarchical representation (or feature) of the input $\boldsymbol{x}$. Here, for a $L$-layer deep network $\psi_{\boldsymbol{\Theta}}(\boldsymbol{x})$, each layer is composed of an affine transformation, followed by a nonlinear activation $\sigma(\cdot)$ (e.g., ReLU) and normalization functions (e.g., BatchNorm Ioffe & Szegedy (2015)).

**Notations for multi-label dataset.** Let $[K] := \{1, 2, \dots, K\}$ denote the set of labels. For each $m \in [K]$, let $\binom{[K]}{m} := \{S \subseteq [K] : |S| = m\}$ denote the set of all subsets of $[K]$ with size $m$. Throughout this work, we consider a fixed multi-label training dataset of the form $\{\boldsymbol{x}_i, \boldsymbol{y}_{S_i}\}_{i=1}^N$, where $N$ is the size of the training set and $S_i$ is a nonempty proper subset of the labels. For instance, $S_i = \{\texttt{cat}\}$ and $S_{i'} = \{\texttt{dog}, \texttt{bird}\}$. Each label $\boldsymbol{y}_{S_i} \in \mathbb{R}^K$ is a *multi-hot-encoding* vector:

$$j\text{-th entry of } \boldsymbol{y}_{S_i} = \begin{cases} 1 & : j \in S_i \\ 0 & : \text{otherwise.} \end{cases} \tag{2}$$

The *Multiplicity* of a training sample $(\boldsymbol{x}_i, \boldsymbol{y}_{S_i})$ is defined as the cardinality of $|S_i|$ of $S_i$, i.e., the number of labels relevant to $\boldsymbol{x}_i$. Additionally, we refer to a feature learned for the sample $(\boldsymbol{x}_i, \boldsymbol{y}_{S_i})$ as the Multiplicity-m feature, if $|S_i| = m$. The Multiplicity-m feature matrix $\boldsymbol{H}_m$ is column-wise comprised of a collection of Multiplicity-m feature vectors. Moreover, we use $M := \max_{i \in [N]} |S_i|$ to denote the largest multiplicity in the training set. Additionally, to distinguish imbalanced class samples between Multiplicities, for each $m \in [M]$, we use $n_m := |\{i \in [N] : |S_i| = m\}|$ to denote the number of samples in each class of a multiplicity order $m$ (or Multiplicity $m$). Note that $M \in \{1, \dots, K-1\}$ in general, and a M-lab problem reduces to M-clf when $M = 1$.

**The "pick-all-labels" loss.** Since M-lab is a generalization of M-clf, recent work Menon et al. (2019) studied various ways of converting a M-clf loss into a M-lab loss, a process referred to as *reduction*.[1] In this work, we analyze the *pick-all-labels* (PAL) method of reducing the cross-entropy (CE) loss to a M-lab loss, which is the *default* option implemented by

---

[1]"Reduction" refers to reformulating M-lab problems in the simpler framework of M-clf problems.

`torch.nn.CrossEntropyLoss` from the deep learning library PyTorch Paszke et al. (2019). The benefit of PAL approach is that the more difficult problem of multi-label can be approached using insights from multi-class learning using well-understood losses such as the cross-entropy, one of the most commonly used loss functions:

$$\mathcal{L}_{\text{CE}}(\boldsymbol{z}, \boldsymbol{y}_k) \; := \; -\log\left(\exp(z_k)/\textstyle\sum_{\ell=1}^K \exp(z_\ell)\right).$$

where $\boldsymbol{z} = \boldsymbol{W}\boldsymbol{h}$ is called the logits, and $\boldsymbol{y}_k$ is the one-hot encoding for the $k$-th class. To convert the CE loss into a `M-clf` loss via the PAL method, for any given label set $S$, consider decomposing a multi-hot label $\boldsymbol{y}_S$ as a summation of one-hot labels: $\boldsymbol{y}_S = \sum_{k\in S} \boldsymbol{y}_k$. Thus, we can define the *pick-all-labels cross-entropy* loss as

$$\mathcal{L}_{\text{PAL-CE}}(\boldsymbol{z}, \boldsymbol{y}_S) \; := \; \textstyle\sum_{k\in S} \mathcal{L}_{\text{CE}}(\boldsymbol{z}, \boldsymbol{y}_k).$$

In this work, we focus exclusively on the CE loss under the PAL framework, below we simply write $\mathcal{L}_{\text{PAL}}$ to denote $\mathcal{L}_{\text{PAL-CE}}$. However, by drawing inspiration from recent research Zhou et al. (2022b), it should be noted that under the PAL framework the phenomenon of `M-lab` NC can be generalized beyond cross-entropy to encompass a variety of other loss functions, such as mean squared error (MSE), label smoothing, focal loss, and potentially a class of Fenchel-Young Losses Blondel et al. (2020). Putting it all together, training deep neural networks for `M-lab` can be stated as follows:

$$\min_{\boldsymbol{\Theta}} \tfrac{1}{N}\textstyle\sum_{i=1}^N \mathcal{L}_{\text{PAL}}(\boldsymbol{W}\phi_{\boldsymbol{\theta}}(\boldsymbol{x}_i) + \boldsymbol{b}, \boldsymbol{y}_{S_i}) + \lambda\|\boldsymbol{\Theta}\|_F^2, \tag{3}$$

where $\boldsymbol{\Theta} = \{\boldsymbol{W}, \boldsymbol{b}, \boldsymbol{\theta}\}$ denote all parameters and $\lambda > 0$ controls the strength of weight decay. Here, weight decay prevents the norm of linear classifier and feature matrix goes to infinity or $0$.

**Optimization under the unconstrained feature model (UFM).**  Analyzing the nonconvex loss in 3 can be notoriously difficult due to the highly non-linear characteristic of the deep network $\phi_{\boldsymbol{\theta}}(\boldsymbol{x}_i)$. In this work, we simplify the study by treating the feature $\boldsymbol{h}_i = \phi_{\boldsymbol{\theta}}(\boldsymbol{x}_i)$ of each input $\boldsymbol{x}_i$ as a *free* optimization variable. More specifically, we study the following problem under UFM:

**Definition 1** (Nonconvex Training Loss under UFM). *Let* $\mathbf{Y} = [\boldsymbol{y}_{S_1} \cdots \boldsymbol{y}_{S_N}] \in \mathbb{R}^{K\times N}$ *be the multi-hot encoding matrix whose $i$-th column is given by the multi-hot vector $\boldsymbol{y}_{S_i} \in \mathbb{R}^K$. We consider the following optimization problem under UFM:*

$$\min_{\boldsymbol{W}, \boldsymbol{H}, \boldsymbol{b}} f(\boldsymbol{W}, \boldsymbol{H}, \boldsymbol{b}) \; := \; g(\boldsymbol{W}\boldsymbol{H} + \boldsymbol{b}, \boldsymbol{Y}) + \lambda_W\|\boldsymbol{W}\|_F^2 + \lambda_H\|\boldsymbol{H}\|_F^2 + \lambda_b\|\boldsymbol{b}\|_2^2 \tag{4}$$

*with the penalty $\lambda_W, \lambda_H, \lambda_b > 0$. Here, the linear classifier $\boldsymbol{W} \in \mathbb{R}^{K\times d}$, the features $\boldsymbol{H} = [\boldsymbol{h}_1, \cdots, \boldsymbol{h}_N] \in \mathbb{R}^{d\times N}$, and the bias $\boldsymbol{b} \in \mathbb{R}^K$ are all unconstrained optimization variables, and we refer to the columns of $\boldsymbol{H}$, denoted $\boldsymbol{h}_i$, as the* unconstrained last layer features *of the input samples $\boldsymbol{x}_i$. Additionally, $g(\cdot)$ is the PAL loss, denoted by*

$$g(\mathbf{WH} + \mathbf{b}, \boldsymbol{Y}) := \tfrac{1}{N}\mathcal{L}_{\text{PAL}}(\mathbf{WH} + \mathbf{b}, \boldsymbol{Y}) \; := \; \tfrac{1}{N}\textstyle\sum_{i=1}^N \mathcal{L}_{\text{PAL}}(\mathbf{W}\boldsymbol{h}_i + \mathbf{b}, \boldsymbol{y}_{S_i}).$$

Analysis of NC under UFM has been extensively studied in recent works Zhu et al. (2021); Fang et al. (2021); Ji et al. (2022); Yaras et al. (2022); Mixon et al. (2022); Zhou et al. (2022a); Tirer & Bruna (2022), the motivation behind the UFM is the fact that modern networks are highly overparameterized and they are universal approximators Cybenko (1989); Zhang et al. (2021). Although the objective function is seemingly a simple extension of `M-clf` case, our work shows that the global optimizers of Problem 4 for `M-lab` substantially differs from that of the `M-clf` that we present in the following.

## 3 MAIN RESULTS

In this section, we rigorously analyze the global geometry of the optimizer of (4) and its nonconvex optimization landscape, and present our main results in Theorem 1 and Theorem 2. For `M-lab`, we show that the global minimizers of Problem 4 exhibit a more generic structure than the vanilla NC in `M-clf` (see Figure 1), where higher multiplicity features are formed by a scaled average of associated Multiplicity-1 features that we introduce in detail below.

### 3.1 MULTI-LABEL NEURAL COLLAPSE (M-LAB NC)

To motivate our theoretical results in the next section, we find experimentally (see Section 4 for details) that an overparameterized neural network trained on a Multiplicity-1 balanced data[2] using the objective (3) to the terminal phase satisfies the properties below which we collectively refer to as **multi-label neural collapse** (`M-lab` NC):

---

[2]Here, theoretically, we allow imbalancedness across different multiplicity. Moreover, empirically we find that `M-lab` NC still holds if training data of high-order multiplicity is imbalanced or even has missing classes.

1. **Variability collapse:** The within-class variability of last-layer features across different multiplicity and different classes all collapses to zero. In other words, the individual features of each class of each multiplicity concentrate to their respective class-means.

2. $(\ast)$ **Convergence to Self-duality of Multiplicity-$1$ features $H_1$ :** The rows of the last-layer linear classifier $W$ and the class means of Multiplicity-1 feature $H$ are collinear, i.e., $h_i^\star \propto w^{\star k}$ when the label set $S_i = \{k\}$ is a singleton set.

3. $(\ast)$ **Convergence to the M−lab ETF:** Multiplicity-1 features $H_1 := \left\{ h_i^\star | i : |S_i| = 1 \right\}$ form a Simplex Equiangular Tight Frame, similar to the M−clf setting Papyan et al. (2020); Fang et al. (2021); Zhu et al. (2021). Moreover, for any higher multiplicity $m > 1$, *the class means for Multiplicity-$m$ features are scaled averages of associated Multiplicity-1 features means over the elements of the corresponding label set*. In other words, $h_i^\star \propto \sum_{k \in S_i} w^{\star k}$ (see the bottom line of Figure 1). This is true regardless of class imbalanced-ness between multiplicities.

**Remarks.** The M−lab NC can be viewed as a more general version of the vanilla NC in M−clf Papyan et al. (2020), where we mark the difference above by a "$(\ast)$". The M−lab ETF implies that, in the pick-all-labels approach to multi-label classification, deep networks learn discriminant and informative features for Multiplicity-1 subset of the training data, and uses them to construct higher multiplicity features as scaled average of associated Multiplicity-1 features. We propose a measure $\mathcal{NC}_m$ to quantify this phenomenon and verify them for practical neural networks in Section 4 below.

Such a result is quite intuitive. For example, consider a sample $i \in [N]$ whose training label $y_{S_i}$ has Multiplicity-2, e.g., $S_i = \{\texttt{cat}, \texttt{dog}\}$. The multi-hot vector label $y_{S_i}$ decomposes as a scaled average of one-hot labels of Multiplicity-1, namely, $y_{S_i} = \sum_{k \in S_i} y_k$. Ideally, the learned representation $h_i^\star$ should satisfy such a property as well: that $h_i^\star$ is a scaled average of several $h_{i'}^\star$'s where each $i' \in [N]$ corresponds to an training instance of Multiplicity-1. The learned representation of an image containing both cat and dog should be a scaled average of the learned representation of images containing only a cat or a dog. Moreover, between multiplicities, the number of samples does *not* need to be balanced. For example, the M−lab NC still holds if there are more training samples for the category $(\texttt{ant}, \texttt{bee})$(Multiplicity-2) than that of $(\texttt{cat}, \texttt{dog}, \texttt{elk})$ (Multiplicity-3).

## 3.2 GLOBAL OPTIMALITY & BENIGN LANDSCAPE UNDER UFM

**Global optimality for M−lab NC.** For M−lab, in the following we show that the M−lab NC is the only global solution to the nonconvex problem in Definition 1. We consider the setting that the training data may exhibit imbalanced-ness between different multiplicities while maintaining class balanced-ness within each multiplicity. For instance, there might be 1000 samples for each class in Multiplicity-1 labels, but only 500 samples for each class within Multiplicity-2 labels, and so forth.

**Theorem 1 (Global Optimality Conditions).** *In the setting of Definition 1, assume the feature dimension is no smaller than number of classes, i.e., $d \geq K - 1$, and assume the data balanced-ness condition above. Then any global optimizer $W^\star, H^\star, b^\star$ of the optimization problem* (4) *satisfies:*

$$w^\star := \|w^{\star 1}\|_2 = \|w^{\star 2}\|_2 = \cdots = \|w^{\star K}\|_2, \quad and \quad b^\star = b^\star \mathbf{1}, \tag{5}$$

*where either $b^\star = 0$ or $\lambda_b = 0$. Moreover, the global minimizer $W^\star, H^\star, b^\star$ satisfies the M−lab NC properties introduced in Section 3.1, in the sense that*

- *The linear classifier matrix $W^{\star\top} \in \mathbb{R}^{d \times K}$ forms a $K$-simplex ETF up to scaling and rotation, i.e., for any $U \in \mathbb{R}^{d \times d}$ s.t. $U^\top U = \mathbf{I}_d$, the rotated and normalized matrix $M := \frac{1}{w^\star} U W^{\star\top}$ satisfies*

$$M^\top M = \frac{K}{K-1} \left( \mathbf{I}_K - \frac{1}{K} \mathbf{1}_K \mathbf{1}_K^\top \right). \tag{6}$$

- *For each feature $h_i^\star$ (i.e., the $i$-th column $h_i^\star$ of $H^\star$) with $i \in [N]$, there exist unique positive real numbers $C_1, C_2, \ldots, C_M > 0$ such that the following holds:*

$$h_i^\star = C_1 w^{\star k} \qquad when\ S_i = \{k\},\ k \in [K], \qquad (Multiplicity = 1\ Case) \tag{7}$$

$$h_i^\star = C_m \textstyle\sum_{k \in S_i} w^{\star k} \quad when\ |S_i| = m,\ 1 < m \leq M. \qquad (Multiplicity > 1\ Case) \tag{8}$$

**Remarks.** While it may appear intuitive and straightforward to extend the analysis of vanilla NC in M−clf to M−lab NC Zhu et al. (2021), the combinatorial nature of high multiplicity features and the interplay between the linear classifier $W$ and these class-imbalanced high multiplicity features present significant challenges for analysis. For instance, previous attempts to prove M−clf NC

utilized Jensen's inequality and the concavity of the logarithmic function, but these methods are not effective for M-lab NC. Instead, we analyze the gradient of the pick-all-labels cross-entropy and leverage its strict convexity to directly construct the desired lower bound. In Appendix A, we offer a more detailed outline of the proof, presenting a lemma-by-lemma comparison with Zhu et al. (2021).

We briefly outline our proofs as follows: essentially, our proof method first breaks down the $g(\boldsymbol{WH} + \boldsymbol{b}, \boldsymbol{Y})$ component of the objective function in (Problem 4) into numerous subproblems $g_m(\boldsymbol{WH}_m + \boldsymbol{b}, \boldsymbol{Y}_m)$, categorized by multiplicity. We determine lower bounds for each $g_m$ and establish the conditions for equality attainment for each multiplicity level. Subsequently, we confirm that these sets of lower bounds for different $m$ values can be attained simultaneously, thus constructing a global optimizer where the overall global objective (4) is reached. We demonstrate that all optimizers can be recovered using this approach. The detailed proof of our results is deferred to Appendix C.

Next, we delve into the interpretation and ramifications of our findings from various perspectives.

- **The global solutions of Problem 4 satisfy M-lab NC.** In the UFM context, our findings imply that every global solution of the loss function (4) exhibits the M-lab NC that we presented in Section 3.1. First, the reduction of feature variability within each class and multiplicity is inferred from Equations 7 and 8. This occurs because all features of the designated class and multiplicity align with the (scaled averages of) linear classifiers, meaning they are equal to their feature means with no variability. Second, the convergence of feature means to the M-lab ETF can be observed from Equation (6), (7), and (8). For Multiplicity-1 features $\boldsymbol{H}_1^\star$, Equation (7) implies that the feature mean $\overline{\boldsymbol{H}}_1^\star$ converges to $\boldsymbol{W}$; this, coupled with Equation (6), implies that the feature means $\overline{\boldsymbol{H}}_1^\star$ of Multiplicity-1 forms a simplex ETF. Moreover, the structure of scaled averages in Equation (8) implies the M-lab ETF for feature means of high multiplicity. Finally, the convergence of Multiplicity-1 features towards self-duality can be deduced from Equation (7).

- **Data imbalanced-ness in M-lab.** Due to the scarcity of higher multiplicity labels in the training set, the imbalanced-ness of training data samples could be a more serious issue in M-lab than M-clf in practice. Recall that there are two types of data imbalanced-ness: (i) the imbalanced-ness between classes *within* each multiplicity and (ii) the imbalanced-ness of classes *among* different multiplicities. Interestingly, as long as Multiplicity-1 training samples remain balanced between classes, our experimental results in Figure 2 and Figure 4 imply that the M-lab NC still holds regardless of both within and among multiplicity imbalanced-ness in higher multiplicity. This demonstrate the practicality of our result, given that achieving balance in Multiplicity-1 sample data is relatively easy. However, if classes of Multiplicity-1 are imbalanced, we suspect more general minority collapse phenomenon would happen Fang et al. (2021); Thrampoulidis et al. (2022), which is worth of further investigation.

- **Scaled average coefficients for M-lab ETF with high multiplicity.** The features of high multiplicity are scaled average of Multiplicity-1 features, and these scaled average coefficients are *simple and structured* as shown in Equation (8). As illustrated in Figure 1 (i.e., $K = 3$, $M = 2$), the feature $\boldsymbol{h}_i^\star$ of Multiplicity-$m$ associated with class-index $S_i$ can be viewed as a *scaled average* of Multiplicity-1 features in the index set $S_i$. Here, the coefficients $\{C_m\}_{m=1}^M$, which are shared across all features of the same multiplicity, could be expressed as

$$C_m = \frac{K-1}{\|\boldsymbol{W}\|_F^2} \log(\frac{K-m}{m} c_{1,m}), \quad \forall m$$

where $\{c_{1,m}\}_{m=1}^M$ exist and they satisfy a set of nonlinear equations.[3]

- **Improving M-lab training via M-lab NC.** As a direct result of our theory shown in Table 1, we can achieve parameter efficient training for M-lab by fixing the last layer classifier as simplex ETF and reducing the feature dimension $d$ to $K$. Furthermore, since higher order multiplicity features are essentially scaled averages of associated Multiplicity-1 features, there is a potential opportunity to design a regularization that encourages features to exhibit the scaled averaging behavior. Such regularization can help constrain the solution space, leading to improved performance or accelerated training process Zhu et al. (2021).

**Nonconvex landscape analysis.** Due to the nonconvex nature of Problem (3), the characterization of global optimality alone in Theorem 1 is not sufficient for guaranteeing efficient optimization

---

[3]please refer to the Appendix C for more details on the nonlinear equations.

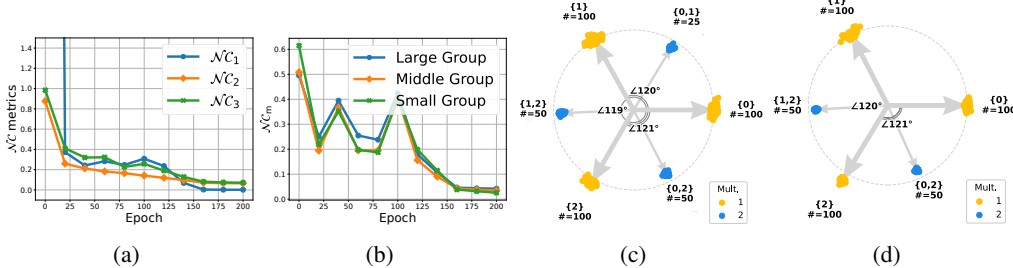

Figure 2: **M-lab NC holds with imbalanced data.** (a) and (b) plot metrics that measures M-lab NC on M-lab Cifar10; (c) and (d) directly visualize learned features on M-lab MNIST, where one multiplicity-2 class is missing in the set up which results in the reduced M-lab NC geometry. More experimental details are deferred to Section 4.

to those desired global solutions. Thus, we further study the global landscape of Problem (3) by characterizing all of its critical points, we show the following result.

**Theorem 2** (**Benign Optimization Landscape**). *Suppose the same setting of Theorem 1, and assume the feature dimension is larger than the number of classes, i.e., $d > K$, and the number of training samples for each class are balanced within each multiplicity. Then the function $f(\boldsymbol{W}, \boldsymbol{H}, \boldsymbol{b})$ in Problem (4) is a strict saddle function with no spurious local minimum in the sense that:*

- *Any local minimizer of $f$ is a global solution of the form described in Theorem 1.*
- *Any critical point $(\boldsymbol{W}, \boldsymbol{H}, \boldsymbol{b})$ of $f$ that is not a global minimizer is a strict saddle point with negative curvatures, in the sense that there exists some direction $(\boldsymbol{\Delta_W}, \boldsymbol{\Delta_H}, \boldsymbol{\delta_b})$ such that the directional Hessian $\nabla^2 f(\boldsymbol{W}, \boldsymbol{H}, \boldsymbol{b})[\boldsymbol{\Delta_W}, \boldsymbol{\Delta_H}, \boldsymbol{\delta_b}] < 0$.*

Because the PAL loss for M-lab is reduced from the CE loss in M-clf, the above result can be generalized from the result in Zhu et al. (2021). We defer detailed proofs to Appendix D.

## 4 EXPERIMENTS

In this section, we conduct a series of experiments to further demonstrate and analyze the M-lab NC on different practical deep networks with various multi-label datasets. First, Figure 3 shows that all practical deep networks exhibit M-lab NC during the terminal phase of training. Second, we investigate M-lab NC under multiplicity imbalanced-ness on both synthetic (Figure 2) and real data (Figure 4), demonstrating that M-lab NC holds irrespective of imbalanced-ness in higher multiplicity data. Finally, we show that achieve significant parameter savings in training deep networks without compromising performance by using M-lab NC. We begin this section by providing an overview of the training datasets and experimental setups.

**Training dataset & experimental setup.** We created synthetic Multi-label MNIST LeCun et al. (2010) and Cifar10 Krizhevsky et al. (2009) datasets by applying zero-padding to each image, increasing its width and height to twice the original size, and then combining it with another padded image from a different class. An illustration of generated multi-label samples can be found in Figure 5. To create the training dataset, for $m = 1$ scenario, we randomly pick 3100 images in each class, and for $m = 2$, we generated 200 images for each combination of classes using the pad-stack method described earlier. Therefore, the total number of images in the training dataset is calculated as $10 \times 3100 + \binom{10}{2} \times 200 = 40000$. For the test dataset, we included 800 images for each class in the $m = 1$ scenario and 50 images for each combination of classes in the $m = 2$ scenario, resulting in a total of 10250 images. To further validate our findings, we conducted additional testing on the practical SVHN dataset (Netzer et al., 2011) alongside the synthetic dataset. In order to preserve the natural characteristics of the SVHN dataset, we applied minimal pre-processing only to ensure a balanced scenario for multiplicity-1, while leaving other aspects of the dataset untouched.

In terms of training deep networks for M-lab, we use standard ResNet He et al. (2016) and VGG Simonyan & Zisserman (2014) network architecture. Throughout all the experiments, we use an SGD optimizer with fixed batch size 128, weight decay $5 \times 10^{-4}$ and momentum 0.9. The learning rate is initially set to $1 \times 10^{-1}$ and dynamically decays to $1 \times 10^{-3}$ following a CosineAnnealing learning rate scheduler as described in Loshchilov & Hutter (2017). The total number of epochs is set to 200 for all experiments.

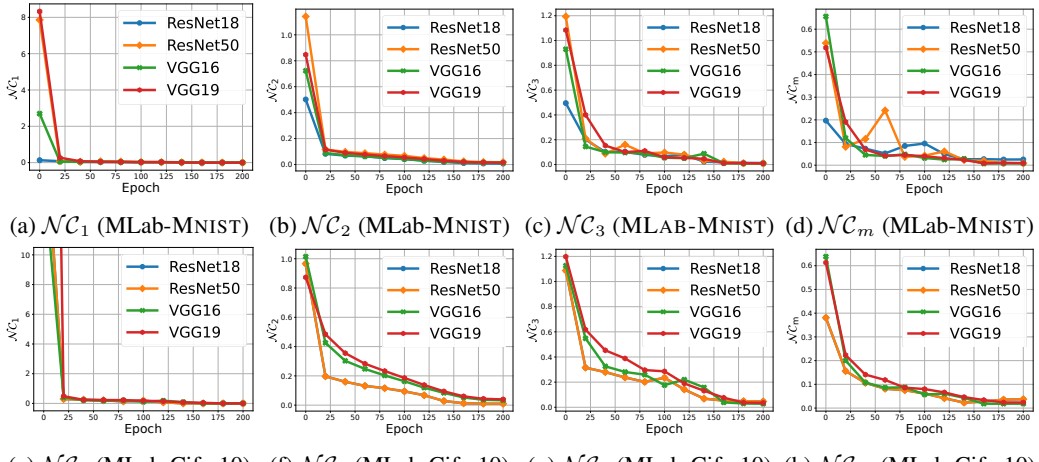

(a) $\mathcal{NC}_1$ (MLab-MNIST)  (b) $\mathcal{NC}_2$ (MLab-MNIST)  (c) $\mathcal{NC}_3$ (MLAB-MNIST)  (d) $\mathcal{NC}_m$ (MLab-MNIST)

(e) $\mathcal{NC}_1$ (MLab-Cifar10)  (f) $\mathcal{NC}_2$ (MLab-Cifar10)  (g) $\mathcal{NC}_3$ (MLab-Cifar10)  (h) $\mathcal{NC}_m$ (MLab-Cifar10)

Figure 3: **Prevalence of `M-lab` NC across different network architectures** on MNIST (top) and Cifar10 (bottom). From the left to the right, the plots show the four metrics, $\mathcal{NC}_1, \mathcal{NC}_2, \mathcal{NC}_3$, and $\mathcal{NC}_m$, for measuring `M-lab` NC.

**Experimental demonstration of `M-lab` NC on practical deep networks.** Based upon the experimental setup, we first demonstrate that `M-lab` NC happens on practical networks trained with `M-lab` datasets, as suggested by our theory. To show this, we need some metrics to measure `M-lab` NC on the last-layer features and classifiers of deep networks.

As showed in Section 3.1, because the original NC in `M-clf` still holds for Multiplicity-1 samples, we use the original metrics $\mathcal{NC}_1$ (measuring the within-class variability collapse), $\mathcal{NC}_2$ (measuring convergence of learned classifier and feature class means to simplex ETF), and $\mathcal{NC}_3$ (measuring the convergence to self-duality) introduced in Papyan et al. (2020) to measure `M-lab` NC on Multiplicity-1 features $\boldsymbol{H}_1$ and classifier $\boldsymbol{W}$. Additionally, we also use the $\mathcal{NC}_1$ metric to measure variability collapse on high multiplicity features $\boldsymbol{H}_m$ $(m > 1)$. Finally, to measure `M-lab` ETF on Multiplicity-2 features,[4] we propose a new angle metric $\mathcal{NC}_m$, which is defined as:

$$\mathcal{NC}_m = \frac{\text{Avg.}(\{geo_\angle(\overline{\boldsymbol{h}}_i, \ \overline{\boldsymbol{h}}_j + \overline{\boldsymbol{h}}_\ell) : |S_i| = 2, |S_j| = |S_\ell| = 1, \ S_i = S_j \cup S_\ell\})}{\text{Avg.}(\{geo_\angle(\overline{\boldsymbol{h}}_{i'}, \ \overline{\boldsymbol{h}}_{j'} + \overline{\boldsymbol{h}}_{\ell'}) : |S_{i'}| = 2, |S_{j'}| = |S_{\ell'}| = 1\})}$$

where $geo\angle$ represents the geometric angle between two vectors and $\overline{\boldsymbol{h}}_i$ is the mean of all features in the label set $S_i$. Intuitively, our $\mathcal{NC}_m$ measures the angle relationship between features means of different label sets or classes. The numerator calculates the average angle difference between multiplicity-2 features means and the sum of their multiplicity-1 component features means. while the denominator serves as a normalization factor that is the average of all existing pairs regardless of the relationship.[5] As training progresses, the numerator will converge to 0, while the denominator becomes larger demonstrating the angle collapsing. As shown in Figure 3 and Figure 4, practical networks do exhibit `M-lab` NC, and such a phenomenon is prevalent across network architectures and datasets. Specifically, the four metrics, evaluated on four different network architectures and two different datasets, all converge to zero as the training progresses towards the terminal phase.

**`M-lab` NC holds with training data imbalanced-ness in high order multiplicity.** Supported and inspired by Theorem 1, where $\boldsymbol{W}$ only collapse to $\boldsymbol{H}_1$, experimentally we found that as long as the training samples of Multiplicity-1 remain balanced, we can still observe `M-lab` NC regardless of the imbalanced-ness in high order multiplicity. To verify this, we create multi-label cifar10 and MNIST datasets. The cifar10 dataset has balanced Multiplicity-1 samples (5000 for each class). For the classes of Multiplicity-2, we divide them into 3 groups: the large group (500 samples), the middle group (50 samples), and the small group (5 samples). We run a ResNet18 model with this dataset and

---

[4]This is because our dataset only contains labels up to Multiplicity-2. The $\mathcal{NC}_m$ could be easily extended to capture scaled average for other higher multiplities

[5]For example, if we have 4 total classes for multiplicity-1 samples, they corresponds to 4 features means and hence 6 different sums if we randomly pick 2 features means to sum up. Multiplicity-2 then have $\binom{4}{2} = 6$ features means, there is in total 36 possible angles to calculate, we averaged these 36 angles and use that as the denominator.

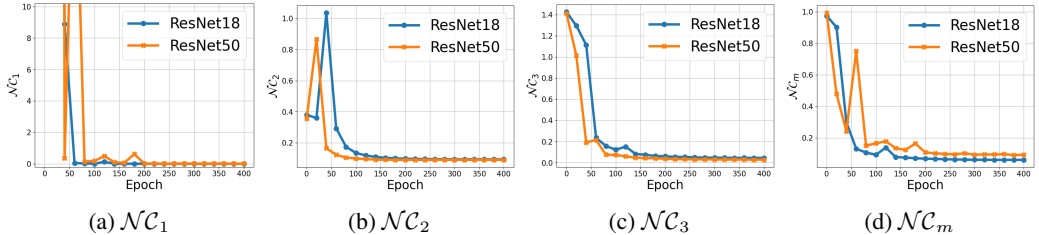

| (a) $\mathcal{NC}_1$ | (b) $\mathcal{NC}_2$ | (c) $\mathcal{NC}_3$ | (d) $\mathcal{NC}_m$ |

Figure 4: **Prevalence of `M-lab` NC on the SVHN dataset**. We train ResNets models on the SVHN dataset Netzer et al. (2011) for $400$ epochs and report $\mathcal{NC}_1, \mathcal{NC}_2, \mathcal{NC}_3$, and $\mathcal{NC}_m$, for measuring `M-lab` NC, respectively.

| Dataset / Arch. | ResNet18 | | ResNet50 | | VGG16 | | VGG19 | |
|---|---|---|---|---|---|---|---|---|
| | Learned | ETF | Learned | ETF | Learned | ETF | Learned | ETF |
| **Test IoU** | | | | | | | | |
| **MLab-MNIST** | 99.47 | 99.37 | 99.43 | 99.42 | 99.47 | 99.50 | 99.45 | 99.49 |
| **MLab-Cifar10** | 87.73 | 87.66 | 88.91 | 88.56 | 86.85 | 87.38 | 86.77 | 86.93 |
| **Percentage of parameter saved** | | | | | | | | |
| **MLab-MNIST** | | | | | | | | |
| **MLab-Cifar10** | 0% | 20.71% | 0% | 4.45% | 0% | 15.75% | 0% | 11.58% |

Table 1: **Comparison of the performances and parameter efficiency between learned and fixed ETF classifier.** When counting parameters, we consider all parameters that require gradient calculation during back-propagation.

report the metrics of measuring `M-lab` NC in Figure 2 (a) (b). We can observe that not only $\mathcal{NC}_1$ to $\mathcal{NC}_3$ collapse to zero, the $\mathcal{NC}_m$ metric is also converging zero for all 3 groups of different size. For Figure 2 (c) (d) on `M-lab` MNIST, we can see from the visualization of the features vectors that the scaled average property still holds despite a missing class in higher multiplicity. Here, we train a simple Convolution plus Multi-layer perceptron model with this dataset. This suggests that `M-lab` NC even under data imbalanced-ness in high order multiplicity.

Besides the synthetic dataset, we also tested on the real dataset SVHN Netzer et al. (2011). We conduct minimal preprocessing to ensure it has balanced Multiplicity-1 samples.[6] Subsequently, we assessed the trends of NC metrics on this dataset, as depicted in Figure 4. The plots affirm the continued validity of our analysis within real-world settings.

**`M-lab` NC guided parameter-efficient training.** With the knowledge of `M-lab` NC in hand, we can make direct modifications to the model architecture to achieve parameter savings without compromising performance for `M-lab`classification. Specifically, parameter saving could come from two folds: (i) given the existence of $\mathcal{NC}$ in the multi-label case with $d \geq K$, we can reduce the dimensionality of the penultimate features to match the number of labels (i.e., we set $d = K$); (ii) recognizing that the final linear classifier will converge to a simplex ETF as the training converges, we can initialize the weight matrix of the classifier as a simplex ETF from the start and refrain from updating it during training. By doing so, our experimental results in Table 1 demonstrate that we can achieve parameter reductions of up to $20\%$ without sacrificing the performance of the model.[7]

## 5 CONCLUSION

In this study, we extensively analyzed the NC phenomenon in `M-lab` Zhu et al. (2021); Fang et al. (2021); Ji et al. (2022). Based upon the UFM, our results establish that `M-lab` ETFs are the only global minimizers of the PAL loss function, incorporating weight decay and bias. These findings hold significant implications for improve the performance and training efficiency of `M-lab` tasks. As a future direction, it would be interesting to investigate `M-lab` NC to improve test performance through better designs of loss functions and regularization techniques Zhou et al. (2022b).

---

[6]For more detail, we refer readers to Figure 6 in the Appendix.

[7]We use intersection over union (IoU) to measure model performances, in `M-lab`, we define $\text{IoU}(\hat{\boldsymbol{y}}, \boldsymbol{y}) = ||\boldsymbol{y}||_0 \cdot (\hat{\boldsymbol{y}}^T \boldsymbol{y}) \in [0, 1]$. Here, the ground truth $\boldsymbol{y}$ represents a probability vector that always sums to 1.

## REPRODUCTIVITY STATEMENT

The complete proof of Theorem 1 and Theorem 2 are shown in Appendix C and Appendix D respectively. Our code is available in the supplementary materiel.

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

# Appendix

APPENDIX ORGANIZATION

In Appendix A, we compare and contrast in details of our work with Zhu et al. (2021). In Appendix B, we illustrate synthetic multi-label MNIST and Cifar10 dataset and show the details of SVHN training data. In Appendix C and Appendix D, we present the proofs for results from the main paper Theorem 1 and Theorem 2, respectively.

## A    DISCUSSION ON RELATIONSHIP TO ZHU ET AL. (2021)

Although our work is inspired by Zhu et al. (2021), our main results as well as the techniques used to establish them significantly depart from that of Zhu et al. (2021). We elaborate on this in the following.

**Technical contribution:** First of all, the proof of the global optimality in the multi-label (M-lab) setting is highly nontrivial, and cannot be simply inferred from Theorem 3.1 in Zhu et al. (2021). The proof of our main result requires a significant amount of new techniques and key Lemmas. The unique challenges of M-lab learning include (1) the combinatorial nature of high multiplicity features, and (2) the interplay between the linear classifier $W$ and these class-imbalanced high multiplicity features. Prior methods, such as those in Zhu et al. (2021) that relied on Jensen's inequality and the concavity of the log function to establish the M-clf NC, fall short in the M-lab scenario. For example, our Lemma 8 leverages novel techniques to address the issue of high-multiplicity samples, which are specific to multi-label problems. We perform a careful calculation of the gradient of the pick-all-labels cross-entropy loss function to formulate a precise lower bound.

**Broader contributions of our work:** Moreover, we posit that our work's contribution extends much beyond the technical aspects, where this is the first work showing the prevalence of a generalized NC phenomenon for multi-label learning both experimentally and theoretically. More surprisingly, our research reveals that the ETF structure remains valid for multiplicity-1 features despite the data imbalance across different multiplicities (as shown in fig. 2). This phenomenon is corroborated by our experimental findings as well as by our theoretical analysis. This insight could lead to potential new pathways for advancing multi-label learning, such as the development of more effective decision-making rules and strategies to manage data imbalances.

In the following, we provide more details on the difference between our proof method for M-lab NC differs and that for M-clf NC (as in Zhu et al. (2021)). The difference primarily due to technical challenges stemming from combinatorial structure of the higher multiplicity data samples in multi-label learning setting. To deal with these challenges, we developed new proving techniques for lower bounds, equality conditions, which are detailed by new probabilistic and matrix theory (Lemma 4, Lemma 5, Lemma 6, Lemma 7). These techniques generalize Zhu et al. (2021)'s proof and implies M-clf NC with only single-multiplicity data.

- To deal with the combinatorial structures of high multiplicity features, our key Lemma 8 (compared with Lemma B.5 in Zhu et al. (2021)), proves a linear lower bound for Pick-All-Label cross-entropy (PAL-CE) loss, which cannot be deduced from the Lemma B.5. More specifically, we prove this linear lower bound for M-lab by a careful analysis of the gradient of the loss directly. Moreover, our result in Lemma 8 is stronger and more general, which implies Lemma B.5 in Zhu et al. (2021) when no high multiplicity samples present. Furthermore, our tightness condition for the lower-bound uncovers an intriguing property which we call the "in-group and out-group" property unique to the M-lab setting.

- To deal with the interplay between linear classifier $W$ and the high multiplicity features, our Lemma 2 (compared to Lemma B.3 in Zhu et al. (2021)) decomposes the loss into different multiplicities, establishing lower bounds for each component and and equality conditions for achieving those lower bounds. In particular, we also showed that these lower bounds can be simultaneously achieved across distinct multiplicities, resulting in a tight global lower

bound. This is highly nontrivial and unique to multi-label learning, which cannot be deduced from Lemma B.3 in Zhu et al. (2021).

- In our Lemma 3 (compared to Lemma B.4 in Zhu et al. (2021)), we characterize the geometry of the multi-label NC. The key departure from Lemma B.4 in Zhu et al. (2021) is that we show that the higher multiplicity feature means converge towards the scaled average of their associated tag feature means, which we call the "scaled-average property". Furthermore, we demonstrate that the associated scaled average coefficient can be determined by solving a system of equations. To obtain theoretical analysis of such scaled average property, we introduce additional Lemma 4, Lemma 5, Lemma 6, and Lemma 7, incorporating a novel probabilistic and matrix analysis technique to comprehensively establish and complete the proof. Due to the unique challenges in the multi-label learning, none of these can be directly deduced from the results in Zhu et al. (2021).

## B DATASET DETAILS

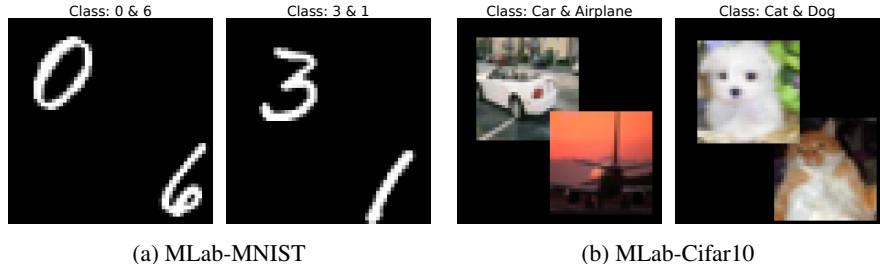

(a) MLab-MNIST  (b) MLab-Cifar10

Figure 5: **Illustration of synthetic multi-label MNIST (left) and Cifar10 (right) datasets.**

The detailed information of SVHN dataset are included in Figure 6

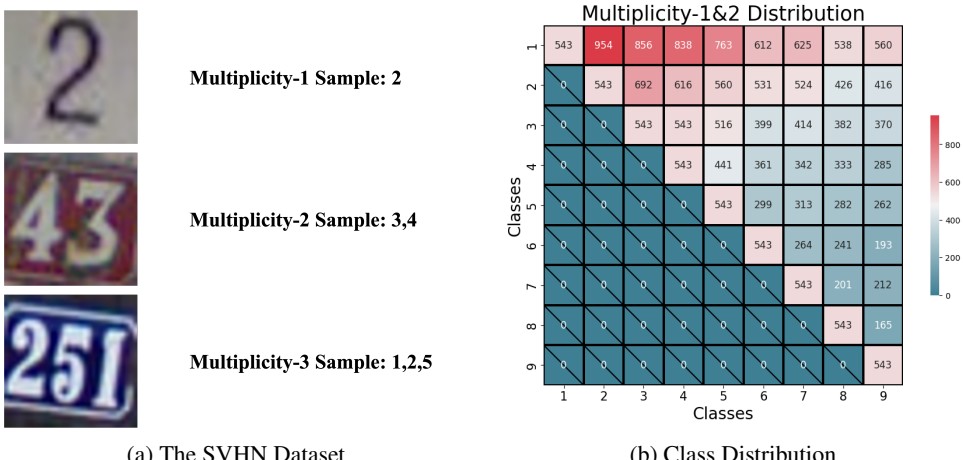

(a) The SVHN Dataset  (b) Class Distribution

Figure 6: **Our usage of the SVHN dataset.** As illustrated in (a), the Street View House Numbers (SVHN) Dataset (Netzer et al., 2011) comprises labeled numerical characters and inherently serves as a multi-label learning dataset. We applied minimal preprocessing to achieve balance specifically within the Multiplicity-1 scenario, as evidenced by the diagonal entries in (b). Furthermore, we omitted samples with Multiplicity-4 and above, as these images posed considerable recognition challenges. Notably, the Multiplicity-2 case remained largely imbalanced, as observed in the off-diagonal entries in (b). Nonetheless, our findings remained robust and consistent in this scenario, as evidenced in Figure 4.

## C  OPTIMALITY CONDITION

The purpose of this section is to prove Theorem 1. As such, throughout this section, we assume that we are in the situation of the statement of said theorem. Due to the additional complexity of the `M-lab` setting compared to the `M-clf` setting, analysis of the `M-lab` NC requires substantially more notations. These notations, which are defined in appendix C.1, while not necessary for *stating* Theorem 1, are crucial for the proofs in appendix C.2 .

### C.1  ADDITIONAL NOTATIONS

For the reader's convenience, we recall the following:

$$N := \text{number of samples} \tag{9}$$

$$N_m := \text{number of samples } i \in [N] \text{ such that } |S_i| = m \tag{10}$$

$$n_m := N_m / \binom{K}{m} \tag{11}$$

$$\binom{[K]}{m} := \{S \subseteq [K] : |S| = m\} \tag{12}$$

$$M := \text{largest } m \text{ such that } n_m \neq 0 \tag{13}$$

$$d := \text{dimension of the last layer features} \tag{14}$$

#### C.1.1  LEXICOGRAPHICAL ORDERING ON SUBSETS

For each $m \leq K$, recall from the above that the set of subsets of $[K]$ of size $m$ is denoted by the commonly used, suggestive notation $\binom{[K]}{m}$. Moreover, $|\binom{[K]}{m}| = \binom{K}{m}$.

▷ **Notation convention**. Assume the *lexicographical ordering* on $\binom{[K]}{m}$. Thus, for each $k \in \binom{K}{m}$, the *k-th subset of* $\binom{[K]}{m}$ is well-defined.

For example, when $K = 5$ and $m = 2$, there are $\binom{5}{2} = 10$ elements in $\binom{[5]}{2}$ which, when listed in the lexicographic ordering, are

$$\underbrace{\{1,2\}}_{\text{1st}}, \underbrace{\{1,3\}}_{\text{2nd}}, \underbrace{\{1,4\}}_{\text{3rd}}, \underbrace{\{1,5\}}_{\text{4th}}, \underbrace{\{2,3\}}_{\cdots}, \{2,4\}, \{2,5\}, \{3,4\}, \{3,5\}, \underbrace{\{4,5\}}_{\text{10th}}.$$

In general, we use the notation $S_{m,k}$ to denote the $k$-th subset of $\binom{[K]}{m}$. In other words,

$$\binom{[K]}{m} = \{S_{m,1}, \ S_{m,2}, \ \ldots, \ S_{m,\binom{K}{m}}\}.$$

#### C.1.2  BLOCK SUBMATRICES OF THE LAST LAYER FEATURE MATRIX

Without the loss of generality, we assume that the sample indices $i \in [N]$ are sorted such that $|S_i|$ is non-decreasing, i.e., $|S_1| \leq \cdots \leq |S_i| \leq \cdots \leq |S_N|$. Clearly, this does not affect the optimization problem itself. Denote the set of indices of Multiplicity-$m$ samples by $\mathcal{I}_m := \{i \in [N] : |S_i| = m\}$. Thus, we have

$$\mathcal{I}_1 = \{1, \ldots, N_1\}, \ \mathcal{I}_2 = \{1 + N_1, \ldots, N_2 + N_1\}, \ \cdots \mathcal{I}_m = \{1 + \sum_{\ell=1}^{m} N_\ell, \ldots, N_m + \sum_{\ell=1}^{m} N_\ell\}, \cdots$$

Below, it will be helpful to define the notation

$$\mathcal{I}_{m,S} := \{i \in [N] : S_i = S\}$$

for each $m = 1, \ldots, M$ and $S \in \binom{[K]}{m}$.

▷ **Notation convention**. Define the block-submatrices $\boldsymbol{H}_1, \ldots, \boldsymbol{H}_M$ of $\boldsymbol{H}$ such that

1. $\boldsymbol{H}_m \in \mathbb{R}^{d \times N_m}$
2. $\boldsymbol{H} = [\boldsymbol{H}_1 \quad \boldsymbol{H}_2 \quad \cdots \quad \boldsymbol{H}_M]$

Thus, as in the main paper, the columns of $\boldsymbol{H}_m$ correspond to the features of $\mathcal{I}_m$.

### C.1.3   DECOMPOSITION OF THE LOSS

Define

$$g_m(\boldsymbol{W}\boldsymbol{H}_m + \boldsymbol{b}, \boldsymbol{Y}) := \frac{1}{N_m} \sum_{i \in \mathcal{I}_m} \underline{\mathcal{L}}_{\text{PAL}}(\boldsymbol{W}\boldsymbol{h}_i + \boldsymbol{b}, \boldsymbol{y}_{S_i}). \tag{15}$$

Intuitively, $g_m$ is the contribution to $g$ from the Multiplicity-$m$ samples. More precisely, the function $g(\boldsymbol{W}\boldsymbol{H} + \boldsymbol{b}, \boldsymbol{Y})$ from Equation (4) can be decomposed as

$$g(\boldsymbol{W}\boldsymbol{H} + \boldsymbol{b}, \boldsymbol{Y}) = \sum_{m=1}^{M} \frac{N_m}{N} g_m(\boldsymbol{W}\boldsymbol{H}_m + \boldsymbol{b}, \boldsymbol{Y}). \tag{16}$$

### C.1.4   TRIPLE INDICES NOTATION

Next, we state precisely the data balanced-ness condition from Theorem 1. In order to state the condition, we need some additional notations. Fix some $m \in \{1, \ldots, M\}$ and let $S \in \binom{[K]}{m}$. Define

$$n_{m,S} := \{i \in [N] : S_i = S\}. \tag{17}$$

Theorem 1 made the following **data balanced-ness condition**:

$$n_{m,S} = N_m / \binom{K}{m} =: n_m \text{ for all } S \in \binom{[K]}{m}. \tag{18}$$

In other words, for a fixed $m \in [M]$, the set $\mathcal{I}_{m,S}$ has the same constant cardinality equal to $n_m$ ranging across all $S \in \binom{[K]}{m}$.

By the data balanced-ness condition, we have for a fixed $m = 1, \ldots, M$ that $\mathcal{I}_{m,S}$ have the same number of elements across all $S \in \binom{[K]}{m}$. Moreover, in our notation, we have $|\mathcal{I}_{m,S}| = n_m$. Below, for each $m = 1, \ldots, M$ and for each $S \in \binom{[K]}{m}$, choose an arbitrary ordering on $\mathcal{I}_{m,S}$ once and for all. Every sample is *uniquely* specified by the following three indices:

1. $m \in [M]$ the sample's multiplicity, i.e., $m = |S|$
2. $k \in \binom{K}{m}$ the index such that $S_{m,k}$ is the label set of the sample,
3. $i \in [n_m]$ such that the sample is the $i$-th element of $\mathcal{I}_{m,S_{m,k}}$.

More concisely, we now introduce the

▷ **Notation convention**. Denote each sample by the triplet

$$(m, k, i) \quad \text{where} \quad m \in [M], \ k \in \binom{K}{m}, \ i \in [n_m]. \tag{19}$$

Below, (19) will be referred to as the **triple indices notation** and every sample will be referred to by its triple indices $(m, k, i)$ instead of the previous single index $i \in [N]$. Accordingly, throughout the appendix, columns of $\boldsymbol{H}$ are expressed as $\boldsymbol{h}_{m,k,i}$ instead of the previous $\boldsymbol{h}_i$, and thus the block submatrix $\boldsymbol{H}_m$ of $\boldsymbol{H}$ can be, without the loss of generality, be written as $\boldsymbol{H}_m = [\boldsymbol{h}_{m,k,i}]_{m \in [M], \ k \in \binom{K}{m}, \ i \in [n_m]}$.

Moreover, in the triple indices notation, Equation (15) can be rewritten as

$$g_m(\boldsymbol{W}\boldsymbol{H}_m + \boldsymbol{b}) = \frac{1}{N_m} \sum_{i=1}^{n_m} \sum_{k=1}^{\binom{K}{m}} \mathcal{L}_{\text{PAL}}(\boldsymbol{W}\boldsymbol{h}_{m,k,i}, \boldsymbol{y}_{S_{m,k}}) \tag{20}$$

### C.2   PROOFS

We will first state the proof of Theorem 1 which depends on several lemmas appearing later in the section. Thus, the proof of Theorem 1 serves as a roadmap for the rest of this section.

*Proof of Theorem 1.* Recall the definition of a coercive function: a function $\varphi : \mathbb{R}^n \to \mathbb{R}$ is said to be coercive if $\lim_{\|x\| \to \infty} \varphi(x) = +\infty$. It is well-known that a coercive function attains its infimum which is a global minimum.

Now, note that the objective function $f(\boldsymbol{W}, \boldsymbol{H}, \boldsymbol{b})$ in Problem (4) is *coercive* due to the weight decay regularizers (the terms $\|\boldsymbol{W}\|_F^2$, $\|\boldsymbol{H}\|_F^2$ and $\|\boldsymbol{b}\|_F^2$) and that the pick-all-labels cross-entropy loss is non-negative. Thus, a global minimizer, denoted below as $(\boldsymbol{W}, \boldsymbol{H}, \boldsymbol{b})$, of Problem (4) exists. By Lemma B.2, we know that any critical point $(\boldsymbol{W}, \boldsymbol{H}, \boldsymbol{b})$ of Problem (4) satisfies

$$\boldsymbol{W}^\top \boldsymbol{W} = \frac{\lambda_{\boldsymbol{H}}}{\lambda_{\boldsymbol{W}}} \boldsymbol{H} \boldsymbol{H}^\top.$$

Let $\rho := \|\boldsymbol{W}\|_F^2$. Thus, $\|\boldsymbol{H}\|_F^2 = \frac{\lambda_{\boldsymbol{W}}}{\lambda_{\boldsymbol{H}}} \rho$

We first provide a lower bound for the PAL cross-entropy term $g(\boldsymbol{W}\boldsymbol{H} + \boldsymbol{b}\mathbf{1}^\top)$ and then show that the lower bound is tight if and only if the parameters are in the form described in Theorem 1. For each $m = 1, \ldots, M$, let $c_{1,m} > 0$ be arbitrary, to be determined below. Now by Lemma 2 and Lemma 8, we have

$$g(\boldsymbol{W}\boldsymbol{H} + \boldsymbol{b}) - \Gamma_2 \geq -\frac{1}{N} \sqrt{\sum_{m=1}^{M} \left( \frac{1}{1 + c_{1,m}} \frac{m}{K - m} \right)^2 \kappa_m n_m \binom{K}{m}^2} \sqrt{\frac{\lambda_{\boldsymbol{W}}}{\lambda_{\boldsymbol{H}}}} \rho$$

where $\Gamma_2 := \sum_{m=1}^{M} c_{2,m}$ and $c_{2,m}$ is as in Lemma 8. Therefore, we have

$$f(\boldsymbol{W}, \boldsymbol{H}, \boldsymbol{b}) = g(\boldsymbol{W}\boldsymbol{H} + \boldsymbol{b}^\top) + \lambda_{\boldsymbol{W}} \|\boldsymbol{W}\|_F^2 + \lambda_{\boldsymbol{H}} \|\boldsymbol{H}\|_F^2 + \lambda_{\boldsymbol{b}} \|\boldsymbol{b}\|_2^2$$

$$\geq -\frac{1}{N} \sqrt{\sum_{m=1}^{M} \left( \frac{1}{1 + c_{1,m}} \frac{m}{K - m} \right)^2 \kappa_m n_m \binom{K}{m}^2} \sqrt{\frac{\lambda_{\boldsymbol{W}}}{\lambda_{\boldsymbol{H}}}} \rho + \Gamma_2 + 2\lambda_{\boldsymbol{W}} \rho + \frac{\lambda_{\boldsymbol{b}}}{2} \|\boldsymbol{b}\|_2^2$$

$$\geq -\frac{1}{N} \sqrt{\sum_{m=1}^{M} \left( \frac{1}{1 + c_{1,m}} \frac{m}{K - m} \right)^2 \kappa_m n_m \binom{K}{m}^2} \sqrt{\frac{\lambda_{\boldsymbol{W}}}{\lambda_{\boldsymbol{H}}}} \rho + \Gamma_2 + 2\lambda_{\boldsymbol{W}} \rho \qquad (21)$$

where the last inequality becomes an equality whenever either $\lambda_{\boldsymbol{b}} = 0$ or $\boldsymbol{b} = \mathbf{0}$. Furthermore, by Lemma 3, we know that the Inequality (21) becomes an equality *if and only if* $(\boldsymbol{W}, \boldsymbol{H}, \boldsymbol{b})$ satisfy the following:

(I)  $\|\boldsymbol{w}^1\|_2 = \|\boldsymbol{w}^2\|_2 = \cdots = \|\boldsymbol{w}^K\|_2$, and $\boldsymbol{b} = b\mathbf{1}$,

(II)  $\dfrac{1}{\binom{K}{m}} \sum_{k=1}^{\binom{K}{m}} \boldsymbol{h}_{m,k,i} = \mathbf{0}$, and $\sqrt{\dfrac{\binom{K-2}{m-1}}{n_m}} \boldsymbol{w}^k = \sum_{\ell : k \in S_{m,\ell}} \boldsymbol{h}_{m,\ell,i}, \forall m \in [M], k \in [K], i \in [n_m]$,

(III)  $\boldsymbol{W}^\top \boldsymbol{W} = \dfrac{\rho}{K - 1} \left( \boldsymbol{I}_K - \dfrac{1}{K} \mathbf{1}_K \mathbf{1}_K^\top \right)$

(IV) There exist unique positive real numbers $C_1, C_2, \ldots, C_M > 0$ such that the following holds:

$$\boldsymbol{h}_{1,k,i} = C_1 \boldsymbol{w}^\ell \qquad \text{when } S_{1,k} = \{\ell\}, \ \ell \in [K], \qquad \text{(Multiplicity = 1 Case)}$$
$$\boldsymbol{h}_{m,k,i} = C_m \sum_{\ell \in S_{m,k}} \boldsymbol{w}^\ell \quad \text{when } m > 1. \qquad \text{(Multiplicity > 1 Case)}$$

Note that condition (IV) is a restatement of Equation (7) and Equation (8). The choice of the $c_{1,m}$'s is given by (V) from Lemma 3. □

**Lemma 1.** *we have:*

$$\boldsymbol{W}^\top \boldsymbol{W} = \frac{\lambda_{\boldsymbol{H}}}{\lambda_{\boldsymbol{W}}} \boldsymbol{H} \boldsymbol{H}^\top \quad and \quad \rho = \|\boldsymbol{W}\|_F^2 = \frac{\lambda_{\boldsymbol{H}}}{\lambda_{\boldsymbol{W}}} \|\boldsymbol{H}\|_F^2$$

*Proof.* The proceeds identically as in given by Zhu et al. (2021) Lemma B.2 and is thus omitted here. □

The following lemma is the generalization of Zhu et al. (2021) Lemma B.3 to the multilabel case for each multiplicity.

**Lemma 2.** *Let $(\boldsymbol{W}, \boldsymbol{H}, \boldsymbol{b})$ be a critical point for the objective $f$ from Problem (4). Let $c_{1,m} > 0$ be arbitrary and let $\gamma_{1,m} := \frac{1}{1+c_{1,m}} \frac{m}{K-m}$. Define $\kappa_m := \left( \frac{K}{m \binom{K}{m}} \right)^2 \binom{K-2}{m-1}$. Then*

$$g(\boldsymbol{WH} + \boldsymbol{b}) - \Gamma_2 \geq -\frac{1}{N} \sqrt{ \sum_{m=1}^{M} \left( \frac{1}{1+c_{1,m}} \frac{m}{K-m} \right)^2 \kappa_m n_m \binom{K}{m}^2 } \sqrt{\frac{\lambda_{\boldsymbol{W}}}{\lambda_{\boldsymbol{H}}}} \rho. \qquad (22)$$

*where $\rho := \|\boldsymbol{W}\|_F^2$, $\Gamma_2 := \sum_{m=1}^{M} c_{2,m}$ and $c_{2,m}$ is as in Lemma 8.*

Note that $\Gamma_2$ depends on $c_{1,1}, c_{1,2}, \ldots, c_{1,M}$ because $c_{2,m}$ depends on $c_{1,m}$ for each $m \in [M]$.

*Proof.* Throughout this proof, let $\boldsymbol{z}_{m,k,i} := \boldsymbol{W}\boldsymbol{h}_{m,k,i} + \boldsymbol{b}$ and choose the same $\gamma_{1,m}, c_{2,m}$ for all $i$ and $k$. The first part of this proof aim to find the lower bound for each $g_m(\boldsymbol{W}, \boldsymbol{H}_m, \boldsymbol{b})$ along with conditions when the bound is tight. The rest of the proof focus on sum up $g_m$ to get Equation (22). Thus, using Equation (20) with the $\boldsymbol{z}_{m,k,i}$'s, we have that $g_m$ can be written as

$$g_m(\boldsymbol{WH}_m + \boldsymbol{b}) = \frac{1}{N_m} \sum_{i=1}^{n_m} \sum_{k=1}^{\binom{K}{m}} \mathcal{L}_{\text{PAL}}(\boldsymbol{z}_{m,k,i}, \boldsymbol{y}_{S_{m,k}}) \qquad (23)$$

By directly applying Lemma 8, the following lower bound holds:

$$N_m g_m(\boldsymbol{WH}_m + \boldsymbol{b}) \geq \gamma_{1,m} \sum_{i=1}^{n_m} \sum_{k=1}^{\binom{K}{m}} \langle \mathbf{1} - \tfrac{K}{m} \mathbb{I}_S, \boldsymbol{W}\boldsymbol{h}_{m,k,i} + \boldsymbol{b} \rangle + N_m c_{2,m}$$

which implies that

$$\gamma_{1,m}^{-1} (g_m(\boldsymbol{WH}_m + \boldsymbol{b}) - c_{2,m})$$
$$\geq \frac{1}{N_m} \sum_{i=1}^{n_m} \sum_{k=1}^{\binom{K}{m}} \langle \mathbf{1} - \tfrac{K}{m} \mathbb{I}_S, \boldsymbol{W}\boldsymbol{h}_{m,k,i} + \boldsymbol{b} \rangle$$
$$= \frac{1}{N_m} \underbrace{\sum_{i=1}^{n_m} \sum_{k=1}^{\binom{K}{m}} \langle \mathbf{1} - \tfrac{K}{m} \mathbb{I}_S, \boldsymbol{W}\boldsymbol{h}_{m,k,i} \rangle}_{(\star)} + \frac{1}{N_m} \underbrace{\sum_{i=1}^{n_m} \sum_{k=1}^{\binom{K}{m}} \langle \mathbf{1} - \tfrac{K}{m} \mathbb{I}_S, \boldsymbol{b} \rangle}_{(\star\star)} \qquad (24)$$

To further simplify the inequality above, we break it down into two parts, namely, the feature part $(\star)$ and the bias part $(\star\star)$ and analyze each of them separately. We first show that the term $(\star\star)$ is equal to zero. To see this, note that

$$(\star\star) = \sum_{k=1}^{\binom{K}{m}} \left( \sum_{j=1}^{K} b_j - \frac{K}{m} \sum_{j' \in S_{m,k}} b_{j'} \right)$$
$$= \sum_{k=1}^{\binom{K}{m}} \sum_{j=1}^{K} b_j - \frac{K}{m} \sum_{k=1}^{\binom{K}{m}} \sum_{j' \in S_{m,k}} b_{j'}$$
$$= K \binom{K}{m} \bar{b} - \frac{K}{m} m \binom{K}{m} \bar{b}$$
$$= 0 \qquad (25)$$

where $\bar{b} = \frac{1}{K} \sum_{j=1}^{K} b_j$ and $\sum_{k=1}^{\binom{K}{m}} \sum_{j=1}^{K} b_j = K \binom{K}{m} \bar{b}$. Thus

$$\sum_{k=1}^{\binom{K}{m}} \sum_{j' \in S_{m,k}} b_{j'} \overset{(\diamondsuit)}{=} \sum_{j=1}^{K} \sum_{k : j \in S_{m,k}} b_j = \sum_{j=1}^{K} b_j \#\{k : j \in S_{m,k}\} = \sum_{j=1}^{K} \binom{K}{m} \frac{m}{K} b_j = m \binom{K}{m} \bar{b}.$$

Note that the equality at ($\diamondsuit$) holds by switching the order of the summation. Now, substituting the result of Equation (25) into the Inequality (24), we have the new lower bound of $g_m$:

$$\gamma_{1,m}^{-1}(g_m(\boldsymbol{W}\boldsymbol{H}_m + \boldsymbol{b}) - c_{2,m}) \geq \frac{1}{N_m} \sum_{i=1}^{n_m} \underbrace{\sum_{k=1}^{\binom{K}{m}} \langle \boldsymbol{1} - \frac{K}{m}\mathbb{I}_S, \boldsymbol{W}\boldsymbol{h}_{m,k,i} \rangle}_{(\star)} \tag{26}$$

and the bound is tight when conditions are met in Lemma 8. To simplify the expression ($\star$) we first distribute the outer layer summation and further simplify it as:

$$(\star) = \sum_{k=1}^{\binom{K}{m}} \sum_{j=1}^{K} \boldsymbol{h}_{m,k,i}^\top \cdot \boldsymbol{w}^j - \frac{K}{m} \sum_{k=1}^{\binom{K}{m}} \sum_{j' \in S_{m,k}} \boldsymbol{h}_{m,k,i}^\top \cdot \boldsymbol{w}^{j'}$$

$$= \sum_{k=1}^{\binom{K}{m}} \sum_{j=1}^{K} \boldsymbol{h}_{m,k,i}^\top \cdot \boldsymbol{w}^j - \frac{K}{m} \sum_{j=1}^{K} \sum_{k':j \in S_{k'}} \boldsymbol{h}_{m,k',i}^\top \boldsymbol{w}^j \tag{27}$$

$$= \sum_{k=1}^{\binom{K}{m}} \sum_{j=1}^{K} \boldsymbol{h}_{m,k,i}^\top \cdot \boldsymbol{w}^j - \frac{K}{m} \sum_{j=1}^{K} \boldsymbol{h}_{m,\{j\},i}^\top \boldsymbol{w}^j$$

$$= \sum_{j=1}^{K} \sum_{k=1}^{\binom{K}{m}} \boldsymbol{h}_{m,k,i}^\top \cdot \boldsymbol{w}^j - \frac{K}{m} \sum_{j=1}^{K} \boldsymbol{h}_{m,\{j\},i}^\top \boldsymbol{w}^j \tag{28}$$

$$= \sum_{j=1}^{K} \left( \sum_{k=1}^{\binom{K}{m}} \boldsymbol{h}_{m,k,i} - \frac{K}{m}\boldsymbol{h}_{m,\{j\},i} \right)^\top \boldsymbol{w}^j$$

$$= \sum_{j=1}^{K} \left( \binom{K}{m}\overline{\boldsymbol{h}}_{m,\bullet,i} - \frac{K}{m}\boldsymbol{h}_{m,\{j\},i} \right)^\top \boldsymbol{w}^j \tag{29}$$

where we let $\boldsymbol{h}_{m,\{j\},i} = \sum_{k:j \in S_{m,k}} \boldsymbol{h}_{m,k,i}$ and $\overline{\boldsymbol{h}}_{m,\bullet,i}$ be the "average" of $\boldsymbol{h}_{m,k,i}$ over all $k \in \binom{K}{m}$ defined as:

$$\overline{\boldsymbol{h}}_{m,\bullet,i} := \frac{1}{\binom{K}{m}} \sum_{k=1}^{\binom{K}{m}} \boldsymbol{h}_{m,k,i}. \tag{30}$$

Similarly to ($\diamondsuit$), the Equations (27) and (28) holds since we only switch the order of summation. Continuing simplification, we substitute the result in Equations (29) and (25) into Inequality (24) we have:

$$\gamma_{1,m}^{-1}(g_m(\boldsymbol{W}\boldsymbol{H}_m + \boldsymbol{b}) - c_{2,m}) \geq \frac{1}{N_m} \sum_{i=1}^{n_m} \sum_{j=1}^{K} \left( \binom{K}{m}\overline{\boldsymbol{h}}_{m,\bullet,i} - \frac{K}{m}\boldsymbol{h}_{m,\{j\},i} \right)^\top \boldsymbol{w}^j$$

$$= \frac{1}{N_m} \sum_{i=1}^{n_m} \sum_{k=1}^{K} \left( \binom{K}{m}\overline{\boldsymbol{h}}_{m,\bullet,i} - \frac{K}{m}\boldsymbol{h}_{m,\{k\},i} \right)^\top \boldsymbol{w}^k$$

$$= \frac{1}{n_m} \sum_{i=1}^{n_m} \sum_{k=1}^{K} \left( \overline{\boldsymbol{h}}_{m,\bullet,i} - \frac{K}{m\binom{K}{m}}\boldsymbol{h}_{m,\{k\},i} \right)^\top \boldsymbol{w}^k$$

Further more, from the AM-GM inequality (e.g., see Lemma A.2 of Zhu et al. (2021)), we know that for any $\boldsymbol{u}, \boldsymbol{v} \in \mathbb{R}^K$ and any $c_{3,m} > 0$,

$$\boldsymbol{u}^\top \boldsymbol{v} \leq \frac{c_{3,m}}{2}\|\boldsymbol{u}\|_2^2 + \frac{1}{2c_{3,m}}\|\boldsymbol{v}\|_2^2 \tag{31}$$

where the above AM-GM inequality becomes an equality when $c_{3,m}\boldsymbol{u} = \boldsymbol{v}$. Thus letting $\boldsymbol{u} = \boldsymbol{w}^k$ and $\boldsymbol{v} = \left(\overline{\boldsymbol{h}}_{m,\bullet,i} - \frac{K}{m\binom{K}{m}}\boldsymbol{h}_{m,\{k\},i}\right)^{\top}$ and applying the AM-GM inequality, we further have:

$$\gamma_{1,m}^{-1}(g_m(\boldsymbol{W}\boldsymbol{H}_m + \boldsymbol{b}) - c_{2,m})$$

$$\geq \frac{1}{n_m}\sum_{i=1}^{n_m}\sum_{k=1}^{K}\left(\overline{\boldsymbol{h}}_{m,\bullet,i} - \frac{K}{m\binom{K}{m}}\boldsymbol{h}_{m,\{k\},i}\right)^{\top}\boldsymbol{w}^k \tag{32}$$

$$\geq \frac{1}{n_m}\sum_{i=1}^{n_m}\sum_{k=1}^{K}\left(-\frac{c_{3,m}}{2}\|\boldsymbol{w}^k\|_2^2 - \frac{1}{2c_{3,m}}\left\|\overline{\boldsymbol{h}}_{m,\bullet,i} - \frac{K}{m\binom{K}{m}}\boldsymbol{h}_{m,\{k\},i}\right\|_2^2\right)$$

$$= \frac{1}{n_m}\sum_{i=1}^{n_m}\sum_{k=1}^{K}-\frac{c_{3,m}}{2}\|\boldsymbol{w}^k\|_2^2 - \frac{1}{n_m}\sum_{i=1}^{n_m}\sum_{k=1}^{K}\frac{1}{2c_{3,m}}\left\|\overline{\boldsymbol{h}}_{m,\bullet,i} - \frac{K}{m\binom{K}{m}}\boldsymbol{h}_{m,\{k\},i}\right\|_2^2$$

$$= -\frac{c_{3,m}}{2}\|\boldsymbol{W}\|_F^2 - \frac{1}{2c_{3,m}n_m}\sum_{i=1}^{n_m}\sum_{k=1}^{K}\left\|\overline{\boldsymbol{h}}_{m,\bullet,i} - \frac{K}{m\binom{K}{m}}\boldsymbol{h}_{m,\{k\},i}\right\|_2^2$$

$$= -\frac{c_{3,m}}{2}\|\boldsymbol{W}\|_F^2 - \frac{1}{2c_{3,m}n_m}\sum_{i=1}^{n_m}\left(K\|\overline{\boldsymbol{h}}_{m,\bullet,i}\|_2^2 + \left(\frac{K}{m\binom{K}{m}}\right)^2\left(\sum_{k=1}^{K}\|\boldsymbol{h}_{m,\{k\},i}\|_2^2\right)\right.$$
$$\left. - 2K\langle\overline{\boldsymbol{h}}_{m,\bullet,i},\ \overline{\boldsymbol{h}}_{m,\bullet,i}\rangle\right)$$

$$= -\frac{c_{3,m}}{2}\|\boldsymbol{W}\|_F^2 - \frac{1}{2c_{3,m}n_m}\sum_{i=1}^{n_m}\left(\left(\frac{K}{m\binom{K}{m}}\right)^2\left(\sum_{k=1}^{K}\|\boldsymbol{h}_{m,\{k\},i}\|_2^2\right) - K\|\overline{\boldsymbol{h}}_{m,\bullet,i}\|_2^2\right)$$

$$= -\frac{c_{3,m}}{2}\|\boldsymbol{W}\|_F^2 - \frac{\left(\frac{K}{m\binom{K}{m}}\right)^2}{2c_{3,m}n_m}\sum_{i=1}^{n_m}\left(\sum_{k=1}^{K}\|\boldsymbol{h}_{m,\{k\},i}\|_2^2 - K\|\overline{\boldsymbol{h}}_{m,\bullet,i}\|_2^2\right)$$

$$\geq -\frac{c_{3,m}}{2}\|\boldsymbol{W}\|_F^2 - \frac{\left(\frac{K}{m\binom{K}{m}}\right)^2}{2c_{3,m}n_m}\sum_{i=1}^{n_m}\sum_{k=1}^{K}\|\boldsymbol{h}_{m,\{k\},i}\|_2^2 \tag{33}$$

$$= -\frac{c_{3,m}}{2}\|\boldsymbol{W}\|_F^2 - \frac{\left(\frac{K}{m\binom{K}{m}}\right)^2}{2c_{3,m}n_m}\left(\|\boldsymbol{H}_m\boldsymbol{D}_m\|_F^2\right) \tag{34}$$

$$= -\frac{c_{3,m}}{2}\|\boldsymbol{W}\|_F^2 - \frac{\left(\frac{K}{m\binom{K}{m}}\right)^2\binom{K-2}{m-1}}{2c_{3,m}n_m}\left(\|\boldsymbol{H}_m\|_F^2\right) \qquad \text{(by Lemma 7)}$$

$$= -\frac{c_{3,m}}{2}\|\boldsymbol{W}\|_F^2 - \frac{\kappa_m}{2c_{3,m}n_m}\left(\|\boldsymbol{H}_m\|_F^2\right),$$

where we let $\boldsymbol{D}_m = \text{diag}(\boldsymbol{Y}_m^{\top},\cdots,\boldsymbol{Y}_m^{\top}) \in \mathbb{R}^{(n_m*\binom{K}{m})\times(n_m*K)}$ and $\boldsymbol{Y}_m \in \mathbb{R}^{K\times\binom{K}{m}}$ is the many-hot label matrix defined as follows[8]:

$$\boldsymbol{Y}_m = \left[\boldsymbol{y}_{S_{m,k}}\right]_{k\in\binom{K}{m}}.$$

The first Inequality (32) is tight whenever conditions mentioned in Lemma 8 are satisfied and the second inequality is tight if and only if

$$c_{3,m}\boldsymbol{w}^k = \left(\frac{K}{m\binom{K}{m}}\boldsymbol{h}_{m,\{k\},i} - \overline{\boldsymbol{h}}_{m,\bullet,i}\right) \quad \forall k \in [K], \quad i \in [n_m]. \tag{35}$$

---

[8]See Appendix C.1.1 for definition of the $S_{m,k}$ notation

Therefore, we have

$$g_m(\boldsymbol{W}\boldsymbol{H}_m + \boldsymbol{b}) - c_{2,m} \geq -\gamma_{1,m}\frac{c_{3,m}}{2}\|\boldsymbol{W}\|_F^2 - \gamma_{1,m}\frac{\kappa_m}{2c_{3,m}n_m}\left(\|\boldsymbol{H}_m\|_F^2\right). \tag{36}$$

The last Inequality (33) achieves its equality if and only if

$$\overline{\boldsymbol{h}}_{m,\bullet,i} = \boldsymbol{0}, \quad \forall i \in [n_m]. \tag{37}$$

Plugging this into (Equation (35)), we have

$$c_{3,m}\boldsymbol{w}^k = \frac{K}{m\binom{K}{m}}\boldsymbol{h}_{m,\{k\},i}$$

$$\implies c_{3,m}^2 = \frac{\left(\frac{K}{m\binom{K}{m}}\right)^2 \sum_{i=1}^n \sum_{k=1}^K \|\boldsymbol{h}_{m,\{k\},i}\|_F^2}{n_m \sum_{k=1}^K \|\boldsymbol{w}^k\|_2^2}$$

$$= \frac{\left(\frac{K}{m\binom{K}{m}}\right)^2 \binom{K-2}{m-1}\|\boldsymbol{H}_m\|_F^2}{n_m\|\boldsymbol{W}\|_F^2}$$

$$= \frac{\kappa_m}{n_m}\frac{\|\boldsymbol{H}_m\|_F^2}{\|\boldsymbol{W}\|_F^2}$$

$$\implies c_{3,m} = \sqrt{\frac{\kappa_m}{n_m}}\frac{\|\boldsymbol{H}_m\|_F}{\|\boldsymbol{W}\|_F}$$

$$\implies c_{3,m}^2 = \frac{\kappa_m}{n_m}\frac{\|\boldsymbol{H}_m\|_F^2}{\|\boldsymbol{W}\|_F^2}.$$

Now, note that by our definition of $\rho$ and Lemma 1, we get

$$\|\boldsymbol{H}\|_F^2 = \frac{\lambda_{\boldsymbol{W}}}{\lambda_{\boldsymbol{H}}}\rho. \tag{38}$$

Recall from the state of the lemma that we defined $\kappa_m := \left(\frac{K/m}{\binom{K}{m}}\right)^2\binom{K-2}{m-1}$ and that $\gamma_{1,m} := \frac{1}{1+c_{1,m}}\frac{m}{K-m}$. Thus, continuing from Inequality (36), we have

$$\gamma_{1,m}^{-1}(g_m(\boldsymbol{W}\boldsymbol{H}_m + \boldsymbol{b}) - c_{2,m}) \geq -\frac{c_{3,m}}{2}\|\boldsymbol{W}\|_F^2 - \frac{\kappa_m}{2c_{3,m}n_m}\|\boldsymbol{H}_m\|_F^2.$$

Next, let $Q > 0$ be an arbitrary constant, to be determined later such that

$$\gamma_{1,m} = \frac{1}{N_m}Qc_{3,m}^{-1}\frac{\|\boldsymbol{H}_m\|_F^2}{\|\boldsymbol{W}\|_F^2}, \qquad \forall m \in \{1,\ldots,M\}. \tag{39}$$

A remark is in order: at this current point in the proof, it is unclear that such a $Q$ exists. However, in Equation (42), we derive an explicit formula for $Q$ such that Equation (39) holds. Now, given Equation (39), we have

$$g_m(\boldsymbol{W}\boldsymbol{H}_m + \boldsymbol{b}) - c_{2,m} \geq \frac{1}{N_m}Q\left(-\frac{1}{2}\|\boldsymbol{H}_m\|_F^2 - \frac{1}{2}\|\boldsymbol{H}_m\|_F^2\right) = -\frac{1}{N_m}Q\|\boldsymbol{H}_m\|_F^2.$$

Let $\Gamma_2 := \sum_{m=1}^M \frac{N_m}{N}c_{2,m}$. Summing the above inequality on both side over $m = 1,\ldots,M$ according to Equation (16), we have

$$g(\boldsymbol{W}\boldsymbol{H} + \boldsymbol{b}) - \Gamma_2 \geq -\frac{1}{N}Q\sum_{m=1}^M \|\boldsymbol{H}_m\|_F^2 = -\frac{1}{N}Q\|\boldsymbol{H}\|_F^2 = -\frac{1}{N}Q\frac{\lambda_{\boldsymbol{W}}}{\lambda_{\boldsymbol{H}}}\rho. \tag{40}$$

where the last equality is due to Equation (38). Now, we derive the expression for $Q$, which earlier we set to be arbitrary. From Equation (39), we have

$$\frac{1}{1+c_{1,m}}\frac{m}{K-m} = \gamma_{1,m} = \frac{1}{N_m}Q\frac{\sqrt{n_m}}{\sqrt{\kappa_m}}\frac{\|\boldsymbol{H}_m\|_F}{\|\boldsymbol{W}\|_F}. \tag{41}$$

Rearranging and using the fact that $N_m = \binom{K}{m} n_m$, we have

$$\binom{K}{m} \frac{1}{1 + c_{1,m}} \frac{m}{K - m} \sqrt{\kappa_m n_m} = Q \frac{\|\boldsymbol{H}_m\|_F}{\|\boldsymbol{W}\|_F}.$$

Squaring both side, we have

$$\left( \frac{1}{1 + c_{1,m}} \frac{m}{K - m} \right)^2 \kappa_m n_m \binom{K}{m}^2 = Q^2 \frac{\|\boldsymbol{H}_m\|_F^2}{\|\boldsymbol{W}\|_F^2}.$$

Summing over $m = 1, \ldots, M$, we have

$$\sum_{m=1}^{M} \left( \frac{1}{1 + c_{1,m}} \frac{m}{K - m} \right)^2 \kappa_m n_m \binom{K}{m}^2 = Q^2 \sum_{m=1}^{M} \frac{\|\boldsymbol{H}_m\|_F^2}{\|\boldsymbol{W}\|_F^2} = Q^2 \frac{\|\boldsymbol{H}\|_F^2}{\|\boldsymbol{W}\|_F^2} = Q^2 \frac{\lambda_{\boldsymbol{W}}}{\lambda_{\boldsymbol{H}}}$$

Thus, we conclude that

$$Q = \sqrt{\frac{\lambda_{\boldsymbol{H}}}{\lambda_{\boldsymbol{W}}}} \sqrt{\sum_{m=1}^{M} \left( \frac{1}{1 + c_{1,m}} \frac{m}{K - m} \right)^2 \kappa_m n_m \binom{K}{m}^2}. \tag{42}$$

Substituting $Q$ into Equation (41), we get

$$\frac{1}{1 + c_{1,m}} \frac{m}{K - m} = \frac{1}{\binom{K}{m} \sqrt{\kappa_m n_m}} \frac{\|\boldsymbol{H}_m\|_F}{\|\boldsymbol{W}\|_F} \sqrt{\frac{\lambda_{\boldsymbol{H}}}{\lambda_{\boldsymbol{W}}}} \sqrt{\sum_{m'=1}^{M} \left( \frac{1}{1 + c_{1,m'}} \frac{m'}{K - m'} \right)^2 \kappa_{m'} n_{m'} \binom{K}{m'}^2}. \tag{43}$$

Finally substituting $Q$ into Equation (40),

$$g(\boldsymbol{W}\boldsymbol{H} + \boldsymbol{b}) - \Gamma_2 \geq -\frac{1}{N} \sqrt{\sum_{m=1}^{M} \left( \frac{1}{1 + c_{1,m}} \frac{m}{K - m} \right)^2 \kappa_m n_m \binom{K}{m}^2} \sqrt{\frac{\lambda_{\boldsymbol{W}}}{\lambda_{\boldsymbol{H}}}} \rho.$$

which concludes the proof. $\qquad\square$

As a sanity check of the validity of Lemma 2, we briefly revisit the M-clf case where $M = 1$. We show that our Lemma 2 recovers Zhu et al. (2021) Lemma B.3 as a special case. Now, from the definition of $\kappa_m$, we have that $\kappa_1 = 1$. Thus, the above expression reduces to simply

$$Q = \sqrt{\frac{\lambda_{\boldsymbol{H}}}{\lambda_{\boldsymbol{W}} n_1}} \frac{1}{1 + c_{1,1}} \frac{1}{K - 1}.$$

The lower bound from Lemma 2 reduces to simply

$$g_1(\boldsymbol{W}\boldsymbol{H}_1 + \boldsymbol{b}) - \gamma_{2,1} \geq -Q \rho \frac{\lambda_{\boldsymbol{W}}}{\lambda_{\boldsymbol{H}}} = -\frac{1}{1 + c_{1,1}} \frac{1}{K - 1} \rho \sqrt{\frac{\lambda_{\boldsymbol{W}}}{\lambda_{\boldsymbol{H}} n_1}}$$

which exactly matches that of Zhu et al. (2021) Lemma B.3.

Next, we show that the lower bound in Inequality (22) is attained if and only if $(\boldsymbol{W}, \boldsymbol{H}, \boldsymbol{b})$ satisfies the following conditions.

**Lemma 3.** *Under the same assumptions of Lemma 2, the lower bound in Inequality (22) is attained for a critical point $(\boldsymbol{W}, \boldsymbol{H}, \boldsymbol{b})$ of Problem (4) if and only if all of the following hold:*

*(I)* $\quad \|\boldsymbol{w}^1\|_2 = \|\boldsymbol{w}^2\|_2 = \cdots = \|\boldsymbol{w}^K\|_2, \quad$ and $\quad \boldsymbol{b} = b\mathbf{1}$,

*(II)* $\quad \dfrac{1}{\binom{K}{m}} \displaystyle\sum_{k=1}^{\binom{K}{m}} \boldsymbol{h}_{m,k,i} = \mathbf{0}, \quad$ and $\quad \sqrt{\dfrac{\binom{K-2}{m-1}}{n_m}} \dfrac{\|\boldsymbol{H}_m\|_F}{\|\boldsymbol{W}\|_F} \boldsymbol{w}^k = \displaystyle\sum_{\ell : k \in S_{m,\ell}} \boldsymbol{h}_{m,\ell,i}, \forall m \in [M], k \in [K], i \in [n_m]$,

*(III)* $\quad \boldsymbol{W}^\top \boldsymbol{W} = \dfrac{\rho}{K - 1} \left( \boldsymbol{I}_K - \dfrac{1}{K} \mathbf{1}_K \mathbf{1}_K^\top \right)$

*(IV)*     *There exist unique positive real numbers $C_1, C_2, \ldots, C_M > 0$ such that the following holds:*

$$\boldsymbol{h}_{1,k,i} = C_1 \boldsymbol{w}^\ell \qquad \text{when } S_{1,k} = \{\ell\}, \ \ell \in [K], \qquad \textit{(Multiplicity} = 1 \textit{ Case)}$$
$$\boldsymbol{h}_{m,k,i} = C_m \textstyle\sum_{\ell \in S_{m,k}} \boldsymbol{w}^\ell \quad \text{when } m > 1. \qquad \textit{(Multiplicity} > 1 \textit{ Case)}$$

*(See Appendix C.1.1 for the notation $S_{m,k}$.)*

*(V)*     *There exists $c_{1,1}, c_{1,2}, \ldots, c_{1,M} > 0$ such that*

$$\frac{1}{1 + c_{1,m}} \frac{m}{K-m} = \frac{1}{\binom{K}{m}\sqrt{\kappa_m n_m}} \frac{\sqrt{\frac{\binom{K}{m} n_m m (K-m)(K-1)}{K}} * log(\frac{K-m}{m} c_{1,m})}{\rho} \cdot$$
$$\sqrt{\frac{\lambda_{\boldsymbol{H}}}{\lambda_{\boldsymbol{W}}}} \sqrt{\sum_{m'=1}^{M} \left(\frac{1}{1 + c_{1,m'}} \frac{m'}{K-m'}\right)^2 \kappa_{m'} n_{m'} \binom{K}{m'}^2}. \tag{44}$$

The proof of Lemma 3 utilizes the conditions in Lemma 8, and the conditions in Equation (35) and Equation (37) during the proof of Lemma 2.

*Proof.* Similar as in the proof of Lemma 2, define $\boldsymbol{h}_{m,\{k\},i} := \sum_{\ell:k \in S_{m,\ell}} \boldsymbol{h}_{m,\ell,i}$ and $\overline{\boldsymbol{h}}_{m,\bullet,i} := \frac{1}{\binom{K}{m}} \sum_{k=1}^{\binom{K}{m}} \boldsymbol{h}_{m,k,i}$. From the proof of Lemma 2, the lower bound is attained whenever the conditions in Equation (35) and Equation (37) hold, which respectively is equivalent to the following:

$$\overline{\boldsymbol{h}}_{m,\bullet,i} = \boldsymbol{0} \qquad \text{and}$$
$$\sqrt{\frac{\binom{K-2}{m-1}}{n_m}} \frac{\|\boldsymbol{H}_m\|_F}{\|\boldsymbol{W}\|_F} \boldsymbol{w}^k = \boldsymbol{h}_{m,\{k\},i}, \forall m \in [M \ k \in [K], i \in [n_m], \tag{45}$$

In particular, the $m = 1$ case further implies

$$\sum_{k=1}^{K} \boldsymbol{w}^k = \boldsymbol{0}.$$

Next, under the condition described in Equation (45), when $m = 1$, if we want Inequality (22) to become an equality, we only need Inequality (32) to become an equality when $m = 1$, which is true if and only if conditions in Lemma 8 holds for $\boldsymbol{z}_{1,k,i} = \boldsymbol{W}\boldsymbol{h}_{1,k,i} \forall i \in [n_m]$ and $\forall k \in [K]$. First let $[\boldsymbol{z}_{1,k,i}]_j = \boldsymbol{h}_{1,k,i}^\top \boldsymbol{w}^j + b_j$, we would have:

$$\sum_{j=1}^{K}[\boldsymbol{z}_{1,k,i}]_j = K\bar{b} \quad \text{and} \quad K[\boldsymbol{z}_{1,k,i}] = c_{3,1}\left(K\|\boldsymbol{w}^k\|_2^2\right) + Kb_k. \tag{46}$$

We pick $\gamma_{1,1} = 1\beta$, where $\beta$ is defined in (60), to be the same for all $k \in [K]$ in multiplicity one, which also means to pick $\frac{1}{\beta} - (K-1)$ to be the same for all $k \in [K]$ within one multiplicity. Note under the first (*in-group equality*) and second *out-group equality* condition in Lemma 8 and utilize

the condition (46), we have

$$
\begin{aligned}
\frac{1}{\beta} - (K-1) &= \frac{(K-1)\exp(z_{out}) + \exp(z_{in})}{\exp(z_{out})} - (K-1) \\
&= (K-1) + \exp(z_{in} - z_{out}) - (K-1) \\
&= \exp(z_{in} - z_{out}) \\
&= \exp\left(\frac{Kz_{in} - z_{in} - (K-1)z_{out}}{K-1}\right) \\
&= \exp\left(\frac{Kz_{in} - \sum_j z_j}{K-1}\right) \\
&= \left(\exp\left(\frac{\sum_j z_j - Kz_{in}}{K-1}\right)\right)^{-1} \\
&= \left(\exp\left(\frac{\sum_j z_j - Kz_k}{K-1}\right)\right)^{-1} \\
&= \exp\left(\frac{K}{K-1}\left(\bar{b} - c_{3,1}\|\boldsymbol{w}^k\|_2^2\right) - b_k\right)^{-1}
\end{aligned}
$$

Since the scalar $\gamma_{1,1}$ is picked the same for one $m$, but the above equality we have

$$
c_{3,1}\|\boldsymbol{w}^k\|_2^2 - b_k = c_{3,1}\|\boldsymbol{w}^\ell\|_2^2 - b_\ell \quad \forall \ell \neq k. \tag{47}
$$

this directly follows after Equation (29) from the proof in Lemma B.4 of Zhu et al. (2021) to conclude all the conditions except the scaled average condition, which we address next. To this end, we use the second condition in (45) which asserts for $m \geq 2$ that:

$$
\sqrt{\frac{n_m}{\binom{K-2}{m-1}}}\frac{\|\boldsymbol{W}\|_F}{\|\boldsymbol{H}_m\|_F}\boldsymbol{h}_{m,\{k\},i} = \boldsymbol{w}^k
$$

$$
\implies \sqrt{\frac{n_1}{\binom{K-2}{1-1}}}\frac{\|\boldsymbol{W}\|_F}{\|\boldsymbol{H}_1\|_F}\boldsymbol{h}_{1,\{k\},i} = \sqrt{n_1}\frac{\|\boldsymbol{W}\|_F}{\|\boldsymbol{H}_1\|_F}\boldsymbol{h}_{1,k,i} = \boldsymbol{w}^k = \sqrt{\frac{n_m}{\binom{K-2}{m-1}}}\frac{\|\boldsymbol{W}\|_F}{\|\boldsymbol{H}_m\|_F}\boldsymbol{h}_{m,\{k\},i}
$$

$$
\implies \boldsymbol{h}_{m,\{k\},i} = \sqrt{\frac{n_1\binom{K-2}{m-1}}{n_m}}\frac{\|\boldsymbol{H}_m\|_F}{\|\boldsymbol{H}_1\|_F}\boldsymbol{h}_{1,k,i} = c_{h,m}\boldsymbol{h}_{1,k,i} \tag{48}
$$

where $c_{h,m} = \sqrt{\frac{n_1\binom{K-2}{m-1}}{n_m}}\frac{\|\boldsymbol{H}_m\|_F}{\|\boldsymbol{H}_1\|_F}$. Let $\widetilde{\boldsymbol{H}}_1$ (resp. $\widetilde{\boldsymbol{H}}_m$) be the block-submatrix corresponding to the first $K$ columns of $\boldsymbol{H}_1$ (resp. first $\binom{K}{m}$ columns of $\boldsymbol{H}_m$). Define $\widetilde{\boldsymbol{Y}}_1$ and $\widetilde{\boldsymbol{Y}}_m$ similarly. Then, Equation (48) can be equivalently stated in the following matrix form:

$$
c_{h,m}\widetilde{\boldsymbol{H}}_1 = \widetilde{\boldsymbol{H}}_m\widetilde{\boldsymbol{Y}}_m^\top
$$

Let $\boldsymbol{P}_m = \widetilde{\boldsymbol{Y}}_m^\top(\widetilde{\boldsymbol{Y}}_m^\top)^\dagger$ be the projection matrix onto the subspace $\widetilde{\boldsymbol{Y}}_m$, then we have

$$
\widetilde{\boldsymbol{H}}_m\boldsymbol{P}_m = \widetilde{\boldsymbol{H}}_m\widetilde{\boldsymbol{Y}}_m^\top(\widetilde{\boldsymbol{Y}}_m^\top)^\dagger = c_{h,m}\widetilde{\boldsymbol{H}}_1(\widetilde{\boldsymbol{Y}}_m^\top)^\dagger,
$$

which simplifies as

$$
\widetilde{\boldsymbol{H}}_m\boldsymbol{P}_m = c_{h,m}\widetilde{\boldsymbol{H}}_1(\widetilde{\boldsymbol{Y}}_m^\top)^\dagger.
$$

Applying Lemma 4 to the LHS and Lemma 6 to the RHS we have

$$
\widetilde{\boldsymbol{H}}_m = c_{h,m}\widetilde{\boldsymbol{H}}_1(\tau_m\widetilde{\boldsymbol{Y}}_m + \eta_m\boldsymbol{\Theta})
$$

$$
\widetilde{\boldsymbol{H}}_m = c_{h,m}\cdot\tau_m\widetilde{\boldsymbol{H}}_1\widetilde{\boldsymbol{Y}}_m
$$

and substituting $\widetilde{\boldsymbol{H}}_1$ using the relationship between $\widetilde{\boldsymbol{H}}_1$ and $\boldsymbol{W}$, namely, $c_{h,1}\cdot(\boldsymbol{W}^\top) = \widetilde{\boldsymbol{H}}_1$, we now have

$$
\widetilde{\boldsymbol{H}}_m = c_{h,m}\cdot\tau_m\cdot c_{1,m}(\boldsymbol{W}^\top\widetilde{\boldsymbol{Y}}_m)
$$

where

$$C_m = c_{h,m} \cdot c_{h,1} \cdot \tau_m$$

$$= \sqrt{\frac{n_1}{n_m \binom{K-2}{m-1}}} \frac{\|\boldsymbol{H}_m\|_F}{\|\boldsymbol{H}_1\|_F} \cdot \sqrt{\frac{1}{n_1}} \frac{\|\boldsymbol{H}_1\|_F}{\|\boldsymbol{W}\|_F}$$

$$= \sqrt{\frac{1}{n_m \binom{K-2}{m-1}}} \frac{\|\boldsymbol{H}_m\|_F}{\|\boldsymbol{W}\|_F}$$

This proves (IV). Finally, to proof (V), following from Equation (43) in the proof of Lemma 2, we only need to further simplify $\|\boldsymbol{H}_m\|_F$.

We first establish a connection the between $\|\boldsymbol{W}\boldsymbol{H}_m\|_F^2$ and $\|\boldsymbol{H}_m\|_F^2$. By definition of Frobenius norm and the last layer classifier $\boldsymbol{W}$ is an ETF with expression $\boldsymbol{W}^\top \boldsymbol{W} = \frac{\rho}{K-1}\left(\boldsymbol{I}_K - \frac{1}{K}\boldsymbol{1}_K \boldsymbol{1}_K^\top\right)$, we have

$$\|\boldsymbol{W}\boldsymbol{H}_m\|_F^2 = tr(\boldsymbol{W}\boldsymbol{H}_m \boldsymbol{H}_m^\top \boldsymbol{W}^\top)$$

$$= \frac{\rho}{K-1} tr(\boldsymbol{H}_m \boldsymbol{H}_m^\top (\boldsymbol{I}_K - \boldsymbol{1}_K \boldsymbol{1}_K^\top))$$

$$= \frac{\rho}{K-1} \|\boldsymbol{H}_m\|_F^2$$

Since variability within feature already collapse at this point, we can express $\|\boldsymbol{W}\boldsymbol{H}_m\|_F^2$ in terms of $z_{m,in}$ and $z_{m,out}$:

$$\|\boldsymbol{W}\boldsymbol{H}_m\|_F^2 = \frac{\rho}{K-1} \|\boldsymbol{H}_m\|_F^2 = \binom{K}{m} n_m (m z_{m,in}^2 + (K-m) z_{m,out}^2).$$

From the second equality we could express $\|\boldsymbol{H}_m\|$ as:

$$\|\boldsymbol{H}_m\|_F = \sqrt{\frac{\binom{K}{m} n_m (K-1)}{\rho} (m z_{m,in}^2 + (K-m) z_{m,out}^2)} \tag{49}$$

Recall from Lemma 8, we have the following equation to express $z_{m,in}$ and $z_{m,out}$

$$z_{in} - z_{m,out} = log(\frac{K-m}{m} c_{1,m}).$$

As column sum of $\boldsymbol{H}_m$ equals to $\boldsymbol{0}$, the column sum of $\boldsymbol{W}\boldsymbol{H}_m$ also equals to $\boldsymbol{0}$ as well. Given the extra constrain of *in-group equality* and *out-group equality* from Lemma 8, it yields:

$$m z_{m,in} + (K-m) z_{m,out} = 0$$

Now we could solve for $z_{m,in}$ and $z_{m,out}$ in terms of $c_{1,m}$

$$z_{m,in} = \frac{K-m}{K} log\left(\frac{K-m}{m} c_{1,m}\right)$$

$$z_{m,out} = -\frac{m}{K} log\left(\frac{K-m}{m} c_{1,m}\right)$$

Substituting above expression for $z_{m,in}$ and $z_{m,out}$ into Equation (49), we have

$$\|\boldsymbol{H}_m\|_F = \sqrt{\frac{\binom{K}{m} n_m m (K-m)(K-1)}{\rho K}} log(\frac{K-m}{m} c_{1,m})$$

Finally, we substituting the above expression of $\|\boldsymbol{H}_m\|_F$ in to Equation (43) and conclude:

$$\frac{1}{1+c_{1,m}} \frac{m}{K-m} = \frac{1}{\binom{K}{m}\sqrt{\kappa_m n_m}} \frac{\sqrt{\frac{\binom{K}{m} n_m m (K-m)(K-1)}{K}} * log(\frac{K-m}{m} c_{1,m})}{\rho} \cdot$$

$$\sqrt{\frac{\lambda_{\boldsymbol{H}}}{\lambda_{\boldsymbol{W}}}} \sqrt{\sum_{m'=1}^{M} \left(\frac{1}{1+c_{1,m'}} \frac{m'}{K-m'}\right)^2 \kappa_{m'} n_{m'} \binom{K}{m'}^2}.$$

Revisiting and combining results from (IV) and (V), we have the scaled-average constant $C_m$ to be

$$C_m = \sqrt{\frac{1}{n_m \binom{K-2}{m-1}}} \frac{\sqrt{\frac{\binom{K}{m} n_m m (K-m)(K-1)}{\rho K}} \log(\frac{K-m}{m} c_{1,m})}{\|W\|_F}$$

$$= \frac{K-1}{\rho} \log(\frac{K-m}{m} c_{1,m})$$

where $c_{1,m}$ is a solution to the system of equation Equation (44). Note that Equation (44) hold for all $m$. Thus, we could construct a system of equation whose variable are $c_{1,1}, \cdots, c_{1,m}$. Even when missing some multiplicity data, we sill have same number of variable $c_{1,m}$ as equations. We numerically verifies that under various of UFM model setting (i.e. different number of class and different number of multiplicities), $c_{1,m}$ does solves the above system of equation. $\square$

**Lemma 4.** *Let let $P_m = \widetilde{Y}_m^\top (\widetilde{Y}_m^\top)^\dagger$ be the projection matrix then we have, $\widetilde{H}_m P_m = \widetilde{H}_m$*

*Proof.* As $P_m$ is a projection matrix, we have that $\|\widetilde{H}_m\|_F^2 = \|\widetilde{H}_m P_m\|_F^2$ if and only if $\widetilde{H}_m = \widetilde{H}_m P_m$. So it is suffice to show that $\|\widetilde{H}_m\|_F^2 = \|\widetilde{H}_m P_m\|_F^2$. We denote $W \widetilde{H}_m P_m$ as the projection solution and by lemma 5 we have that

$$W \widetilde{H}_m P_m = W \widetilde{H}_m,$$

which further implies that the projection solution $W \widetilde{H}_m P_m$ also solves $g$

$$g(W \widetilde{H}_m, \widetilde{Y}) = g(W \widetilde{H}_m P_m, \widetilde{Y}).$$

When it comes to the regularization term, by minimum norm projection property, we have $\|\widetilde{H}_m\|_F^2 \geq \|\widetilde{H}_m P_m\|_F^2$. Note if the projection solution results in a strictly smaller frobenious norm i.e. $\|\widetilde{H}_m\|_F^2 > \|\widetilde{H}_m P_m\|_F^2$, then $f(W, \widetilde{H}_m P_m, b) < f(W, \widetilde{H}_m, b)$, this contradict the assumption that $Z_m = W \widetilde{H}_m$ is the global solutions of $f$. Thus, the only possible outcomes is that $\|\widetilde{H}_m\|_F^2 = \|\widetilde{H}_m P_m\|_F^2$, which complete the proof. $\square$

**Lemma 5.** *We want to show that the optimal global solution of $f$, $W \widetilde{H}_m P_m$, is the same after projected on to the space of $\widetilde{Y}_m$, i.e., $W \widetilde{H}_m P_m = W \widetilde{H}_m$*

*Proof.* Let $Z_m = W \widetilde{H}_m$ denote the global minimizer of the loss function $f$ for an arbitrary multiplicity $m$. Since $Z_m$ has both the in-group and out-group equality property, we could express it as

$$Z_m = d_1 \widetilde{Y}_m + d_2 \Theta,$$

for some constant $d_1$, $d_2$, and all-one matrix $\Theta$ of proper dimension. Note that it is suffice to show that $Z_m$ lives in the subspace of which the projection matrix $P_m$ projects onto. By lemma 6, as $(\widetilde{Y}_m^\top)^\dagger$ is the Moore–Penrose pseudo-inverse of $\widetilde{Y}_m^\top$ by, we could rewrite $P_m$ as

$$P_m = \widetilde{Y}_m^\top (\widetilde{Y}_m^\top)^\dagger$$
$$= \widetilde{Y}_m^\top \left( \widetilde{Y}_m \widetilde{Y}_m^\top \right)^\dagger \widetilde{Y}_m$$
$$= \widetilde{Y}_m^\top \left( \widetilde{Y}_m \widetilde{Y}_m^\top \right)^{-1} \widetilde{Y}_m.$$

Hence we can see that the subspace which $P_m$ projects onto is spanned by columns/rows of $\widetilde{Y}_m$. In order to show that $Z_m = d_1 \widetilde{Y}_m + d_2 \Theta$ is in the subspace spanned by columns of $\widetilde{Y}_m$, it is suffice to see that the columns sum of $\widetilde{Y}_m = \frac{m}{K} \binom{K}{m} \mathbf{1}$. Thus, we finished the proof. $\square$

**Lemma 6.** *The Moore-Penrose pseudo-inverse of $\widetilde{Y}_m^\top$ has the form $(\widetilde{Y}_m^\top)^\dagger = \tau_m \widetilde{Y}_m + \eta_m \Theta$, where $\Theta$ is the all-one matrix with proper dimension and $\tau_m = \frac{a+c}{bc}$, $\eta_m = -\frac{a}{bc}$, for $a = \frac{m-1}{k-1} \binom{K-1}{m-1}$, $b = \frac{m}{k} \binom{K}{m}$, $c = \frac{m}{k-1} \binom{K-1}{m}$.*

*Proof.* First, we have the column sum of $\widetilde{\boldsymbol{Y}}_m$ can be written as a constant times an all-one vector

$$\sum_j^{\binom{K}{m}} (\widetilde{\boldsymbol{Y}}_m)_{:,j} = \frac{m}{K}\binom{K}{m}\mathbf{1} \tag{50}$$

This property could be seen from a probabilistic perspective. We let $i \in [K]$ be fixed and deterministic, and let $S \subseteq [K]$ be a random subset of size $m$ generating by sampling without replacement. Then

$$Pr\{i \notin S\} = \frac{K-1}{K} \times \frac{K-2}{K-1} \times \cdots \times \frac{K-m}{K-m+1} = \frac{K-m}{K}.$$

This implies that $Pr\{i \in S\} = \frac{m}{K}$ and each entry of the column sum result is exactly $\frac{m}{K}\binom{K}{m}$ as we sum up all $\binom{K}{m}$ columns of $\widetilde{\boldsymbol{Y}}_m$.

Second, the label matrix $\widetilde{\boldsymbol{Y}}_m$ has the property that

$$\widetilde{\boldsymbol{Y}}_m\widetilde{\boldsymbol{Y}}_m^\top = \begin{bmatrix} b & & a \\ & \ddots & \\ a & & b \end{bmatrix}, \quad \widetilde{\boldsymbol{Y}}_m(\boldsymbol{\Theta} - \widetilde{\boldsymbol{Y}}_m^\top) = \begin{bmatrix} 0 & & c \\ & \ddots & \\ c & & 0 \end{bmatrix}, \tag{51}$$

where $a = \frac{m-1}{k-1}\binom{K-1}{m-1}$, $b = \frac{m}{k}\binom{K}{m}$, $c = \frac{m}{k-1}\binom{K-1}{m}$. Again, from a probabilistic perspective, any off-diagonal entry of the product $\widetilde{\boldsymbol{Y}}_m\widetilde{\boldsymbol{Y}}_m^\top$ is equal to $(\widetilde{\boldsymbol{Y}}_m)_{i,:}(\widetilde{\boldsymbol{Y}}_m)_{i',:}^\top$, for $i \neq i'$. Note that $(\widetilde{\boldsymbol{Y}}_m)_{i,:}$ is a row vector of length $\binom{K}{m}$, whose entry are either 0 or 1 and the results of $(\widetilde{\boldsymbol{Y}}_m)_{i,:}(\widetilde{\boldsymbol{Y}}_m)_{i',:}^\top$ would only increase by one if both $(\widetilde{\boldsymbol{Y}}_m)_{i,j} = 1$ and $(\widetilde{\boldsymbol{Y}}_m)_{i',j}^\top = 1$ for $j \in [\binom{K}{m}]$. From the previous property we know that there is $\frac{m}{K}$ probability that $(\widetilde{\boldsymbol{Y}}_m)_{i,j} = 1$. In addition, conditioned on $(\widetilde{\boldsymbol{Y}}_m)_{i,j} = 1$, there are $\frac{m-1}{K-1}$ probability that $(\widetilde{\boldsymbol{Y}}_m)_{i',j} = 1$. Thus, $a = \frac{m}{K}\frac{m-1}{K-1}\binom{K}{m} = \frac{m-1}{K-1}\binom{K-1}{m-1}$. For similar reasoning, we can see that conditioned on $(\widetilde{\boldsymbol{Y}}_m)_{i,j} = 1$, there are $1 - \frac{m-1}{K-1} = \frac{K-m}{K-1}$ probability that $(\boldsymbol{\Theta}_{i',j} - (\widetilde{\boldsymbol{Y}}_m)_{i',j}) = 1$. Thus, $c = \frac{m}{K}\frac{K-m}{K-1}\binom{K}{m} = \frac{m}{K-1}\binom{K-1}{m}$. For the similar probabilistic argument, it is easy to see that diagonal of $\widetilde{\boldsymbol{Y}}_m\widetilde{\boldsymbol{Y}}_m^\top$ are all $b = \frac{m}{K}\binom{K}{m}$ and diagonal of $\widetilde{\boldsymbol{Y}}_m(\boldsymbol{\Theta} - \widetilde{\boldsymbol{Y}}_m^\top)$ are all 0. Then by the second property (Equation (51)), we are about to cook up a left inverse of $\widetilde{\boldsymbol{Y}}^\top$:

$$\frac{1}{b}\left(\widetilde{\boldsymbol{Y}}\widetilde{\boldsymbol{Y}}^\top - \frac{a}{c}(\widetilde{\boldsymbol{Y}}(\boldsymbol{\Theta} - \widetilde{\boldsymbol{Y}}^\top))\right) = \boldsymbol{I}$$

$$\widetilde{\boldsymbol{Y}}\left(\frac{1}{b}\widetilde{\boldsymbol{Y}}^\top - \frac{a}{bc}\boldsymbol{\Theta} + \frac{a}{bc}\widetilde{\boldsymbol{Y}}^\top\right) = \boldsymbol{I}$$

$$\widetilde{\boldsymbol{Y}}\left(\frac{a+c}{bc}\widetilde{\boldsymbol{Y}}^\top - \frac{a}{bc}\boldsymbol{\Theta}\right) = \boldsymbol{I}$$

$$\left(\frac{a+c}{bc}\widetilde{\boldsymbol{Y}} - \frac{a}{bc}\boldsymbol{\Theta}\right)\widetilde{\boldsymbol{Y}}^\top = \boldsymbol{I}$$

Let, $\tau_m = \frac{a+c}{bc}$, $\eta_m = -\frac{a}{bc}$, then the pseudo-inverse of $\widetilde{\boldsymbol{Y}}^\top$, namely $(\widetilde{\boldsymbol{Y}}^\top)^\dagger$ could be written as

$$(\widetilde{\boldsymbol{Y}}^\top)^\dagger = \tau_m\widetilde{\boldsymbol{Y}} + \eta_m\boldsymbol{\Theta}$$

This inverse is also the Moore–Penrose inverse which is unique since it satisfies that:

$$\widetilde{\boldsymbol{Y}}^\top(\widetilde{\boldsymbol{Y}}^\top)^\dagger\widetilde{\boldsymbol{Y}}^\top = \widetilde{\boldsymbol{Y}}^\top\boldsymbol{I} = \widetilde{\boldsymbol{Y}}^\top \tag{52}$$

$$(\widetilde{\boldsymbol{Y}}^\top)^\dagger\widetilde{\boldsymbol{Y}}^\top(\widetilde{\boldsymbol{Y}}^\top)^\dagger = \boldsymbol{I}(\widetilde{\boldsymbol{Y}}^\top)^\dagger = (\widetilde{\boldsymbol{Y}}^\top)^\dagger \tag{53}$$

$$(\widetilde{\boldsymbol{Y}}^\top(\widetilde{\boldsymbol{Y}}^\top)^\dagger)^\top = \widetilde{\boldsymbol{Y}}^\top(\widetilde{\boldsymbol{Y}}^\top)^\dagger \tag{54}$$

$$((\widetilde{\boldsymbol{Y}}^\top)^\dagger\widetilde{\boldsymbol{Y}}^\top)^\top = (\widetilde{\boldsymbol{Y}}^\top)^\dagger\widetilde{\boldsymbol{Y}}^\top \tag{55}$$

$\square$

**Lemma 7.** *We would like to show the following equation holds:*

$$\|\boldsymbol{H}_m \boldsymbol{D}_m\|_F^2 = \binom{K-2}{m-1} \|\boldsymbol{H}_m\|_F^2$$

*Proof.* Note due to how we construct $\boldsymbol{D}_m$, it is suffice to show that $\|\widetilde{\boldsymbol{H}}_m \widetilde{\boldsymbol{Y}}_m^\top\|_F^2 = \binom{K-2}{m-1}\|\widetilde{\boldsymbol{H}}_m\|_F^2$. Recall the definition that $a = \frac{m-1}{k-1}\binom{K-1}{m-1}$ and $b = \frac{m}{k}\binom{K}{m}$. By unwinding the definition of binomial coefficient and simplifying factorial expressions, we can see that $b - a = \binom{K-2}{m-1}$. Along with the assumption that columns sum of $\widetilde{\boldsymbol{H}}_m$ is $\boldsymbol{0}$ i.e. $\overline{\boldsymbol{h}}_{m,\bullet,i} = \boldsymbol{0}, \quad \forall i \in [n_m]$ and the property described in Equation (51), we have

$$\|\widetilde{\boldsymbol{H}}_m \widetilde{\boldsymbol{Y}}_m^\top\|_F^2 = \binom{K-2}{m-1}\|\widetilde{\boldsymbol{H}}_m\|_F^2$$
$$\iff \|\tau_m \widetilde{\boldsymbol{H}}_1 \widetilde{\boldsymbol{Y}}_m \widetilde{\boldsymbol{Y}}_m^\top\|_F^2 = \binom{K-2}{m-1}\|\tau_m \widetilde{\boldsymbol{H}}_1 \widetilde{\boldsymbol{Y}}_m\|_F^2$$
$$\iff \tau_m^2 (b-a)^2 \|\widetilde{\boldsymbol{H}}_1\|_F^2 = \tau_m^2 (b-a)\|\widetilde{\boldsymbol{H}}_1 \widetilde{\boldsymbol{Y}}_m\|_F^2$$
$$\iff (b-a)\|\widetilde{\boldsymbol{H}}_1\|_F^2 = \|\widetilde{\boldsymbol{H}}_1 \widetilde{\boldsymbol{Y}}_m\|_F^2$$
$$\iff (b-a)\|\widetilde{\boldsymbol{H}}_1\|_F^2 = Tr(\widetilde{\boldsymbol{H}}_1 \widetilde{\boldsymbol{Y}}_m \widetilde{\boldsymbol{Y}}_m \widetilde{\boldsymbol{H}}_1^\top)$$
$$\iff (b-a)\|\widetilde{\boldsymbol{H}}_1\|_F^2 = Tr((b-a)\widetilde{\boldsymbol{H}}_1 \widetilde{\boldsymbol{H}}_1^\top)$$
$$\iff (b-a)\|\widetilde{\boldsymbol{H}}_1\|_F^2 = (b-a)\|\widetilde{\boldsymbol{H}}_1\|_F^2$$

Thus, we complete the proof. $\square$

The following result is a `M-lab` generalization of Lemma B.5 from Zhu et al. (2021):

**Lemma 8.** *Let $S \subseteq \{1, \ldots, K\}$ be a subset of size $m$ where $1 \leq m < K$. Then for all $\boldsymbol{z} = (z_1, \ldots, z_K)^\top \in \mathbb{R}^K$ and all $c_{1,m} > 0$, there exists a constant $c_{2,m}$ such that*

$$\mathcal{L}_{\text{PAL}}(\boldsymbol{z}, \boldsymbol{y}_S) \geq \frac{1}{1 + c_{1,m}} \frac{m}{K - m} \cdot \langle \boldsymbol{1} - \tfrac{K}{m}\mathbb{I}_S, \boldsymbol{z}\rangle + c_{2,m}. \tag{56}$$

*In fact, we have*

$$c_{2,m} := \frac{c_{1,m}m}{c_{1,m}+1}\log(m) + \frac{mc_{1,m}}{1+c_{1,m}}\log\left(\frac{c_{1,m}+1}{c_{1,m}}\right) + \frac{m}{c_{1,m}+1}\log\left((K-m)(c_{1,m}+1)\right).$$

*The Inequality (56) is tight, i.e., achieves equality, if and only if $\boldsymbol{z}$ satisfies all of the following:*

1. *For all $i, j \in S$, we have $z_i = z_j$ (in-group equality). Let $z_{\text{in}} \in \mathbb{R}$ denote this constant.*

2. *For all for all $i, j \in S^c$, we have $z_i = z_j$ (out-group equality). Let $z_{\text{out}} \in \mathbb{R}$ denote this constant.*

3. *$z_{\text{in}} - z_{\text{out}} = \log\left(\frac{(K-m)}{m}c_{1,m}\right) = \log\left(\gamma_{1,m}^{-1} - \frac{(K-m)}{m}\right).$*

*Proof.* Let $\boldsymbol{z}$ and $c_{1,m}$ be fixed. For convenience, let $\gamma_{1,m} := \frac{1}{1+c_{1,m}}\frac{m}{K-m}$. Below, let $z_{\text{in}}, z_{\text{out}} \in \mathbb{R}$ be arbitrary to be chosen later. Define $\boldsymbol{z}^* = (z_1^*, \ldots, z_K^*) \in \mathbb{R}^K$ such that

$$z_k^* = \begin{cases} z_{\text{in}} & : k \in S \\ z_{\text{out}} & : k \in S^c. \end{cases} \tag{57}$$

For any $\boldsymbol{z} \in \mathbb{R}^K$, recall from the definition of pick-all-labels cross-entropy loss that

$$\mathcal{L}_{\text{PAL}}(\boldsymbol{z}, \boldsymbol{y}_S) = \sum_{k \in S} \mathcal{L}_{\text{CE}}(\boldsymbol{z}, \boldsymbol{y}_k)$$

In particular, the function $\boldsymbol{z} \mapsto \mathcal{L}_{\text{PAL}}(\boldsymbol{z}, \boldsymbol{y}_S)$ is a sum of strictly convex functions and is itself also strictly convex. Thus, the first order Taylor approximation of $\mathcal{L}_{\text{PAL}}(\boldsymbol{z}, \boldsymbol{y}_S)$ around $\boldsymbol{z}^*$ yields the following lower bound:

$$\mathcal{L}_{\text{PAL}}(\boldsymbol{z}, \boldsymbol{y}_S) \geq \mathcal{L}_{\text{PAL}}(\boldsymbol{z}^*, \boldsymbol{y}_S) + \langle \nabla \mathcal{L}_{\text{PAL}}(\boldsymbol{z}^*, \boldsymbol{y}_S), \ \boldsymbol{z} - \boldsymbol{z}^* \rangle$$
$$= \langle \nabla \mathcal{L}_{\text{PAL}}(\boldsymbol{z}^*, \boldsymbol{y}_S), \ \boldsymbol{z} \rangle + \mathcal{L}_{\text{PAL}}(\boldsymbol{z}^*, \boldsymbol{y}_S) - \langle \nabla \mathcal{L}_{\text{PAL}}(\boldsymbol{z}^*, \boldsymbol{y}_S), \ \boldsymbol{z}^* \rangle \qquad (58)$$

Next, we calculate $\nabla \mathcal{L}_{\text{PAL}}(\boldsymbol{z}^*, \boldsymbol{y}_S)$. First, we observe that

$$\nabla \mathcal{L}_{\text{PAL}}(\boldsymbol{z}^*, \boldsymbol{y}_S) = \sum_{k \in S} \nabla \mathcal{L}_{\text{CE}}(\boldsymbol{z}^*, \boldsymbol{y}_k).$$

Recall the well-known fact that the gradient of the cross-entropy is given by

$$\nabla \mathcal{L}_{\text{CE}}(\boldsymbol{z}^*, \boldsymbol{y}_k) = \text{softmax}(\boldsymbol{z}^*) - \boldsymbol{y}_k. \qquad (59)$$

Below, it is useful to define

$$\alpha := \frac{\exp(z^*_{\text{in}})}{\sum_j \exp(z^*_j)} \quad \text{and} \quad \beta := \frac{\exp(z^*_{\text{out}})}{\sum_j \exp(z^*_j)} \qquad (60)$$

where $\sum_j \exp(z^*_j) = m\exp(z^*_{\text{in}}) + (K-m)\exp(z^*_{\text{out}})$. In view of this notation and the definition of $\boldsymbol{z}^*$ in Equation (57), we have

$$\text{softmax}(\boldsymbol{z}^*) = \alpha \mathbb{I}_S + \beta \mathbb{I}_{S^c} \qquad (61)$$

where we recall that $\mathbb{I}_S$ and $\mathbb{I}_{S^c} \in \mathbb{R}^K$ are the indicator vectors for the set $S$ and $S^c$, respectively. Thus, combining Equation (59) and Equation (61), we get

$$\nabla \mathcal{L}_{\text{PAL}}(\boldsymbol{z}^*, \boldsymbol{y}_S) = \sum_{k \in S} \nabla \mathcal{L}_{\text{CE}}(\boldsymbol{z}^*, \boldsymbol{y}_k) = \sum_{k \in S} (\alpha \mathbb{I}_S + \beta \mathbb{I}_{S^c} - \boldsymbol{y}_k) = m(\alpha \mathbb{I}_S + \beta \mathbb{I}_{S^c}) - \mathbb{I}_S.$$

The above right-hand-side can be rewritten as

$$m(\alpha \mathbb{I}_S + \beta \mathbb{I}_{S^c}) - \mathbb{I}_S = (m\alpha - 1) \cdot \mathbb{I}_S + m\beta \cdot \mathbb{I}_{S^c}$$
$$= (m\alpha - 1 + m\beta - m\beta) \cdot \mathbb{I}_S + m\beta \cdot \mathbb{I}_{S^c}$$
$$= m\beta \cdot \mathbf{1} - (m\beta + 1 - m\alpha) \cdot \mathbb{I}_S$$
$$= m\beta \cdot \left( \mathbf{1} - \frac{m\beta + 1 - m\alpha}{m\beta} \cdot \mathbb{I}_S \right).$$

Note that from Equation (61) we have $m\alpha + (K-m)\beta = 1$. Manipulating this expression algebraically, we have

$$m\alpha + (K-m)\beta = 1$$
$$\iff k - m = \frac{1 - m\alpha}{\beta}$$
$$\iff \frac{1}{\beta}\left( \frac{1}{m} - \alpha \right) = \frac{K}{m} - 1$$
$$\iff 1 + \frac{1}{m\beta} - \frac{\alpha}{\beta} = \frac{K}{m}$$
$$\iff \frac{m\beta + 1 - m\alpha}{m\beta} = \frac{K}{m}.$$

Putting it all together, we have

$$\nabla \mathcal{L}_{\text{PAL}}(\boldsymbol{z}^*, \boldsymbol{y}_S) = m\beta \cdot (\mathbf{1} - \tfrac{K}{m} \cdot \mathbb{I}_S).$$

Thus, combining Equation (58) with the above identity, we have

$$\mathcal{L}_{\text{PAL}}(\boldsymbol{z}, \boldsymbol{y}_S) \geq m\beta \cdot \langle \mathbf{1} - \tfrac{K}{m} \cdot \mathbb{I}_S, \ \boldsymbol{z} \rangle + \mathcal{L}_{\text{PAL}}(\boldsymbol{z}^*, \boldsymbol{y}_S) - m\beta \cdot \langle \mathbf{1} - \tfrac{K}{m} \cdot \mathbb{I}_S, \boldsymbol{z}^* \rangle. \qquad (62)$$

Let

$$c_{2,m} := \mathcal{L}_{\text{PAL}}(\boldsymbol{z}^*, \boldsymbol{y}_S) - m\beta \cdot \langle \mathbf{1} - \tfrac{K}{m} \cdot \mathbb{I}_S, \boldsymbol{z}^* \rangle \qquad (63)$$

Note that this definition depends on $\beta$, which in terms depends in $z_{\text{in}}^*$ and $z_{\text{out}}^*$ which we have not yet defined. To define these quantities, note that in order to derive Equation (56) from Equation (62), a sufficient condition is to ensure that

$$\frac{1}{1+c_{1,m}}\frac{m}{K-m} = m\beta = \frac{m\exp(z_{\text{out}}^*)}{\sum_j \exp(z_j^*)} = \frac{1}{\exp(z_{\text{in}}^* - z_{\text{out}}^*) + \frac{(K-m)}{m}} \tag{64}$$

Rearranging, the above can be rewritten as

$$(1+c_{1,m})\tfrac{K-m}{m} = \exp(z_{\text{in}}^* - z_{\text{out}}^*) + \tfrac{(K-m)}{m} \iff c_{1,m} = \tfrac{m}{K-m}\exp(z_{\text{in}}^* - z_{\text{out}}^*)$$

or, equivalently, as

$$z_{\text{in}}^* - z_{\text{out}}^* = \log\left(\tfrac{(K-m)}{m}c_{1,m}\right). \tag{65}$$

Thus, if we choose $z_{\text{in}}^*, z_{\text{out}}^*$ such that the above holds, then Equation (56) holds.

Finally, we compute the closed-form expression for $c_{2,m}$ defined in Equation (63), which we restate below for convenience:

$$c_{2,m} := \mathcal{L}_{\text{PAL}}(\boldsymbol{z}^*, \boldsymbol{y}_S) - m\beta \cdot \langle \mathbf{1} - \tfrac{K}{m}\cdot\mathbb{I}_S, \boldsymbol{z}^*\rangle$$

The expression for $m\beta$ is given at Equation (64). Moreover, we have

$$\langle \mathbf{1} - \tfrac{K}{m}\cdot\mathbb{I}_S, \boldsymbol{z}^*\rangle = mz_{\text{in}} + (K-m)z_{\text{out}} - \tfrac{K}{m}mz_{\text{in}} = -(K-m)(z_{\text{in}} - z_{\text{out}}).$$

Thus, we have

$$\begin{aligned}
-m\beta \cdot \langle \mathbf{1} - \tfrac{K}{m}\cdot\mathbb{I}_S, \boldsymbol{z}^*\rangle &= \frac{(K-m)(z_{\text{in}} - z_{\text{out}})}{\exp(z_{\text{in}}^* - z_{\text{out}}^*) + \frac{(K-m)}{m}} \\
&= \frac{(K-m)\log\left(\frac{(K-m)}{m}c_{1,m}\right)}{\frac{(K-m)}{m}c_{1,m} + \frac{(K-m)}{m}} \\
&= \frac{m}{c_{1,m}+1}\log\left(\tfrac{(K-m)}{m}c_{1,m}\right).
\end{aligned}$$

On the other hand,

$$\mathcal{L}_{\text{PAL}}(\boldsymbol{z}^*, \boldsymbol{y}_S) = \sum_{k\in S}\mathcal{L}_{\text{CE}}(\boldsymbol{z}^*, \boldsymbol{y}_k)$$

Now,

$$\begin{aligned}
\mathcal{L}_{\text{CE}}(\boldsymbol{z}^*, \boldsymbol{y}_k) &= -\log([\text{softmax}(\boldsymbol{z}^*)]_k) \\
&= -\log(\exp(z_{\text{in}}^*)/(m\exp(z_{\text{in}}^*) + (K-m)\exp(z_{\text{out}}^*))) \\
&= \log(m + (K-m)\exp(z_{\text{out}}^* - z_{\text{in}}^*)) \\
&= \log\left(m + (K-m)(1/\exp(z_{\text{in}}^* - z_{\text{out}}^*))\right) \\
&= \log\left(m + (K-m)\frac{1}{\frac{(K-m)}{m}c_{1,m}}\right) \qquad \text{by Equation (65)} \\
&= \log\left(m + m\frac{1}{c_{1,m}}\right) \\
&= \log\left(m\left(\frac{c_{1,m}+1}{c_{1,m}}\right)\right)
\end{aligned}$$

Thus

$$\mathcal{L}_{\text{PAL}}(\boldsymbol{z}^*, \boldsymbol{y}_S) = m\log\left(m\left(\frac{c_{1,m}+1}{c_{1,m}}\right)\right).$$

Putting it all together, we have

$$c_{2,m} = m\log\left(m\left(\frac{c_{1,m}+1}{c_{1,m}}\right)\right) + \frac{m}{c_{1,m}+1}\log\left(\tfrac{(K-m)}{m}c_{1,m}\right).$$

$$m \log \left( m \left( \frac{c_{1,m} + 1}{c_{1,m}} \right) \right) = m \log(m) + m \log \left( \frac{c_{1,m} + 1}{c_{1,m}} \right)$$

$$\frac{m}{c_{1,m} + 1} \log \left( \frac{(K-m)}{m} c_{1,m} \right) = \frac{m}{c_{1,m} + 1} \log \left( (K-m) c_{1,m} \right) - \frac{m}{c_{1,m} + 1} \log(m)$$

Putting it all together, we have

$$c_{2,m} = m \log \left( m \left( \frac{c_{1,m} + 1}{c_{1,m}} \right) \right) + \frac{m}{c_{1,m} + 1} \log \left( \frac{(K-m)}{m} c_{1,m} \right) \tag{66}$$

$$= m \log(m) + m \log \left( \frac{c_{1,m} + 1}{c_{1,m}} \right) + \frac{m}{c_{1,m} + 1} \log \left( (K-m) c_{1,m} \right) - \frac{m}{c_{1,m} + 1} \log(m) \tag{67}$$

$$= \frac{c_{1,m} m}{c_{1,m} + 1} \log(m) + m \log \left( \frac{c_{1,m} + 1}{c_{1,m}} \right) + \frac{m}{c_{1,m} + 1} \log \left( (K-m) c_{1,m} \right) \tag{68}$$

Next, for simplicity, let us drop the subscript and simply write $c := c_{1,m}$. Then

$$m \log \left( \frac{c+1}{c} \right) + \frac{m}{c+1} \log \left( (K-m)c \right)$$

$$= \frac{m}{1+c} \log \left( \frac{c+1}{c} \right) + \frac{mc}{1+c} \log \left( \frac{c+1}{c} \right) + \frac{m}{c+1} \log \left( (K-m)c \right) \qquad \because \frac{1}{1+c} + \frac{c}{1+c} = 1$$

$$= \frac{m}{1+c} \log \left( \frac{c+1}{c} \right) + \frac{mc}{1+c} \log \left( \frac{c+1}{c} \right) + \frac{m}{c+1} \log \left( (K-m)c \right)$$

$$\quad + \frac{m}{c+1} \log \left( (K-m)(c+1) \right) - \frac{m}{c+1} \log \left( (K-m)(c+1) \right) \qquad \because \text{add a "zero"}$$

$$= \frac{m}{1+c} \log \left( \frac{c+1}{c} \right) + \frac{mc}{1+c} \log \left( \frac{c+1}{c} \right) + \frac{m}{c+1} \log \left( \frac{c}{c+1} \right) \qquad \because \text{property of log}$$

$$\quad + \frac{m}{c+1} \log \left( (K-m)(c+1) \right)$$

$$= \frac{mc}{1+c} \log \left( \frac{c+1}{c} \right) + \frac{m}{c+1} \log \left( (K-m)(c+1) \right) \qquad \because \log(\frac{c+1}{c}) = -\log(\frac{c}{c+1})$$

To conclude, we have

$$c_{2,m} = \frac{c_{1,m} m}{c_{1,m} + 1} \log(m) + \frac{m c_{1,m}}{1 + c_{1,m}} \log \left( \frac{c_{1,m} + 1}{c_{1,m}} \right) + \frac{m}{c_{1,m} + 1} \log \left( (K-m)(c_{1,m} + 1) \right)$$

as desired. □

# D  GLOBAL LANDSCAPE

**Theorem 3** (No Spurious Local Minima and Strict Saddle Property (Generalization of Zhu et al. (2021) Theorem 3.2). *Assume the feature dimension $d > K$, the following function*

$$\min_{\boldsymbol{W}, \boldsymbol{H}, \boldsymbol{b}} f(\boldsymbol{W}, \boldsymbol{H}, \boldsymbol{b}) = \frac{1}{N} \sum_{m=1}^{m} \sum_{i=1}^{n_m} \sum_{k=1}^{\binom{K}{m}} \mathcal{L}_{\texttt{PAL}}(\mathbf{W}\mathbf{h}_{m,k,i} + \mathbf{b}, \mathbf{y}_{S_{m,k}})$$

$$+ \lambda_{\boldsymbol{W}} ||\boldsymbol{W}||_F^2 + \lambda_{\boldsymbol{H}} ||\boldsymbol{H}||_F^2 + \lambda_{\boldsymbol{b}} ||\boldsymbol{b}||_2^2 \tag{69}$$

*with respect to $\boldsymbol{W} \in \mathbb{R}^{K \times d}$, $\boldsymbol{H} = [\boldsymbol{H}_1, ..., \boldsymbol{H}_m] \in \mathbb{R}^{d \times Nm}$ and $\boldsymbol{b} \in \mathbb{R}^K$ is a strict saddle function Ge et al. (2015); Sun et al. (2015); Zhang et al. (2020b) with the following properties:*

- *Any local minimizer of eq. (69) is a global minimizer of the form as shown in Theorem 1*

- *Any critical point of eq. (69) is either a local minimum or has at least one negative curvature direction, i.e., the Hessian $\nabla^2 f(\boldsymbol{W}, \boldsymbol{H}, \boldsymbol{b})$ at this point has at least one negative eigenvalue*

$$\lambda_i(\nabla^2 f(\boldsymbol{W}, \boldsymbol{H}, \boldsymbol{b})) < 0.$$

*Proof of Theorem 3.* We note that the proof for Theorem 3.2 in Zhu et al. (2021) could be directly extended in our analysis. More specifically, the proof in Zhu et al. (2021) relies on a connection for the original loss function to its convex counterpart, in particular, letting $\boldsymbol{Z} = \boldsymbol{W}\boldsymbol{H} \in \mathbb{R}^{K \times N}$ with $N = \sum_m n_m$ and $\alpha = \frac{\lambda_H}{\lambda_W}$, the original proof first shows the following fact:

$$
\begin{aligned}
\min_{\boldsymbol{H}\boldsymbol{W}=\boldsymbol{Z}} \; \lambda_{\boldsymbol{W}}||\boldsymbol{W}||_F^2 + \lambda_{\boldsymbol{H}}||\boldsymbol{H}||_F^2 \; &= \; \sqrt{\lambda_{\boldsymbol{W}}\lambda_{\boldsymbol{H}}} \min_{\boldsymbol{H}\boldsymbol{W}=\boldsymbol{Z}} \; \frac{1}{\sqrt{\alpha}}(||\boldsymbol{W}||_F^2 + \alpha||\boldsymbol{H}||_F^2) \\
&= \; \sqrt{\lambda_{\boldsymbol{W}}\lambda_{\boldsymbol{H}}}||\boldsymbol{Z}||_*.
\end{aligned}
$$

With the above result, the original proof relates the original loss function

$$
\min_{\boldsymbol{W},\boldsymbol{H},\boldsymbol{b}} \; f(\boldsymbol{W},\boldsymbol{H},\boldsymbol{b}) \; := \; g(\boldsymbol{W}\boldsymbol{H} + \boldsymbol{b}\boldsymbol{1}^\top) \; + \; \lambda_{\boldsymbol{W}}||\boldsymbol{W}||_F^2 + \lambda_{\boldsymbol{H}}||\boldsymbol{H}||_F^2 + \lambda_{\boldsymbol{b}}||\boldsymbol{b}||_2^2
$$

with

$$
g(\boldsymbol{W}\boldsymbol{H} + \boldsymbol{b}\boldsymbol{1}^\top) := \frac{1}{N} \sum_{k=1}^{K} \sum_{i=1}^{n} \mathcal{L}_{\text{CE}}(\boldsymbol{W}\boldsymbol{h}_{k,i} + \boldsymbol{b}, \boldsymbol{y}_k),
$$

to a convex problem:

$$
\min_{\boldsymbol{Z} \in \mathbb{R}^{K \times N}, \; \boldsymbol{b} \in \mathbb{R}^K} \; \widetilde{f}(\boldsymbol{Z},\boldsymbol{b}) \; := \; g(\boldsymbol{Z} + \boldsymbol{b}\boldsymbol{1}^\top) + \sqrt{\lambda_{\boldsymbol{W}}\lambda_{\boldsymbol{H}}}||\boldsymbol{Z}||_* + \lambda_{\boldsymbol{b}}||\boldsymbol{b}||_2^2.
$$

In our analysis, by letting $\widetilde{g}(\boldsymbol{W}\boldsymbol{H} + \boldsymbol{b}\boldsymbol{1}^\top) := \frac{1}{Nm} \sum_{m=1}^{m} \sum_{i=1}^{n_m} \sum_{k=1}^{\binom{K}{m}} \mathcal{L}_{\text{PAL}}(\mathbf{W}\mathbf{h}_{m,k,i} + \mathbf{b}, \mathbf{y}_{S_{m,k}})$, we can directly apply the original proof for our problem. For more details, we refer readers to the proof of Theorem 3.2 in Zhu et al. (2021). $\square$

