# OpenReview forum: "A Geometric Analysis of Multi-label Learning under Pick-all-label Loss via Neural Collapse"
_ICLR.cc/2024/Conference — ICLR 2024 Conference Withdrawn Submission_

### Official Review · Reviewer_V285 · 2023-10-30

**Soundness:** 2 fair
**Presentation:** 4 excellent
**Contribution:** 1 poor
**Rating:** 3
**Confidence:** 4

**Summary:**

This paper investigates the training of deep neural networks for multi-label learning through the lens of neural collapse. Its main contributions   are summarized as:

a. This paper shows that the last-layer features and classifier learned via overparameterized deep networks exhibit a more general version of neural collapse.

b. This paper studies the global optimality of a commonly used pick-all-label loss for M-lab and proves that the optimization landscape has benign strict saddle properties so that global solutions can be efficiently achieved.

**Strengths:**

a. This paper is well-written and easy to follow.

b.  I appreciate that this paper provides extensive experiments.

c. Interesting findings. This paper shows that the last-layer features and classifier learned via overparameterized deep networks exhibit a more general version of NC. The high-order Multiplicity features are scaled average of their associated features in Multiplicity-1.

**Weaknesses:**

My main concern is the novelty of results. The main results Theorems 1 and 2 are so similar with the reference [1], i.e. Theorem 1 corresponds to Theorem 3.1 of [1] and Theorem 2 corresponds to Theorem 3.2 of [1]. In my opinion, these results are the extended versions of [1] with a few improvements for multi label learning. It is OK to leverage them, but they are not enough to be the main contributions in this paper.

[1] Zhihui Zhu, Tianyu Ding, Jinxin Zhou, Xiao Li, Chong You, Jeremias Sulam, and Qing Qu. A geometric analysis of neural collapse with unconstrained features. Advances in Neural Information Processing Systems, 34:29820–29834, 2021.

**Questions:**

a. Please discuss more about the results of reference [1]

[1] Zhihui Zhu, Tianyu Ding, Jinxin Zhou, Xiao Li, Chong You, Jeremias Sulam, and Qing Qu. A geometric analysis of neural collapse with unconstrained features. Advances in Neural Information Processing Systems, 34:29820–29834, 2021.

---

> ### Author Response · Authors · 2023-11-15
> **Rebuttal by Authors**
>
> We appreciate that the reviewer finds our work well-presented, interesting, and supported with comprehensive experiments. We address the reviewer’s major concerns in the following:
>
> $\textbf{Q1:}$ "My main concern is the novelty of results. The main results of Theorems 1 and 2 are so similar to the reference [1], i.e. Theorem 1 corresponds to Theorem 3.1 of [1] and Theorem 2 corresponds to Theorem 3.1 of [1]. In my opinion, these results are the extended versions of [1] with a few improvements for multi-label learning. It is OK to leverage them, but they are not enough to be the main contributions in this paper."
>
> $\textbf{A1:}$  Although our work is inspired by [1], our main results as well as the techniques used to establish them significantly depart from that of [1]. We elaborate on this in the following.
>
> * $\textbf{Technical contribution.}$ First of all, the proof of the global optimality in the multi-label (M-lab) setting is highly nontrivial, and cannot be simply inferred from Theorem 3.1 in [1]. The proof of our main result requires a significant amount of new techniques and key Lemmas. This is elaborated in our 16-page long detailed proof in Appendix C. As discussed at the bottom of Page 5 in our paper, the unique challenges of M-lab learning include (1) the combinatorial nature of high multiplicity features, and (2) the interplay between the linear classifier $\boldsymbol{W}$ and these class-imbalanced high multiplicity features. Prior methods, such as those in [1] that relied on Jensen's inequality and the concavity of the log function to establish the multi-class (M-clf) NC, fall short in the M-lab scenario. Our Lemma 8 leverages novel techniques to address the issue of high-multiplicity samples, which are specific to multi-label problems. We perform a careful calculation of the gradient of the pick-all-labels cross-entropy loss function to formulate a precise lower bound. We discuss more detail in A2.
>
> * $\textbf{Broader contributions of our work.}$ Moreover, we posit that our work’s contribution extends much beyond the technical aspects, where this is the first work showing the prevalence of a generalized NC phenomenon for multi-label learning both experimentally and theoretically. More surprisingly, our research reveals that the ETF structure remains valid for multiplicity-1 features despite the data imbalance across different multiplicities (as shown in Figure 2). This phenomenon is corroborated by our experimental findings as well as by our theoretical analysis. This insight could lead to potential new pathways for advancing multi-label learning, such as the development of more effective decision-making rules and strategies to manage data imbalances.
>
> In the revision of our paper, we have made this clear by adding the reference of Zhu et al'21 in the remark below Theorem 1, and added an extra section in Appendix A to discuss the differences between our result and Zhu et al'21 in greater detail. All the updates made are marked in blue.

---

> ### Author Response · Authors · 2023-11-15
> **Rebuttal by Authors**
>
> $\textbf{Q2:}$ “Please discuss more about the results of reference [1]”
>
> $\textbf{A2:}$ In the following, we provide more details on the difference between our proof method for M-lab NC and that for M-clf NC (as in [1]). The difference primarily due to technical challenges stemming from the combinatorial structure of the higher multiplicity data samples in  multi-label learning setting. To deal with these challenges, we developed new proving techniques for lower bounds, and equality conditions, which are detailed by new probabilistic and matrix theory (Lemmas 4, 5, 6, 7). These techniques generalize [1]'s proof and imply M-clf NC with only single-multiplicity data. In the revised paper,we provide a detailed comparison of Lemmas, comparing both [1]’s and our approaches (i.e. M-clf NC and M-lab NC), as shown in the Appendix A of our new revised PDF, highlighted in blue. In the following, we elaborate in more detail.
>
> * To deal with the combinatorial structures of high multiplicity features, our key Lemma 8 (compared with Lemma B.5 in [1]), proves a linear lower bound for Pick-All-Label cross-entropy (PAL-CE) loss, which cannot be deduced from Lemma B.5 of [1]. More specifically, we prove this linear lower bound for M-lab by a careful analysis of the gradient of the loss directly. Moreover, our result in Lemma 8 is stronger and more general, which implies  Lemma B.5 in [1] when no high multiplicity samples are present. Furthermore, our tightness condition for the lower-bound uncovers an intriguing property which we call the “in-group and out-group” property unique to the M-lab setting.
>
> * To deal with the interplay between linear classifier $\\boldsymbol{W}$ and the high multiplicity features, our Lemma 2 (compared to Lemma B.3 in [1]) decomposes the loss into different multiplicities, establishing lower bounds for each component and equality conditions for achieving those lower bounds. In particular, we also showed that these lower bounds can be simultaneously achieved across distinct multiplicities, resulting in a tight global lower bound. This is highly nontrivial and unique to multi-label learning, which cannot be deduced from Lemma B.3 in [1].
>
> * In our Lemma 3 (compared to Lemma B.4 in [1]), we characterize the geometry of the multi-label  NC. The key departure from Lemma B.4 in[1] is that we show that the higher multiplicity feature means converge towards the scaled average of their associated tag feature means, which we call the “scaled-average property”. Furthermore, we demonstrate that the associated scaled average coefficient can be determined by solving a system of equations. To obtain a theoretical analysis of such scaled average property, we introduce additional Lemmas 4, 5, 6, and 7, incorporating a novel probabilistic and matrix analysis technique to comprehensively establish and complete the proof. Due to the unique challenges in multi-label learning, none of these can be directly deduced from the results in [1].

---

> > ### Author Response · Authors · 2023-11-18
> > **Looking forward to your feedback**
> >
> > Dear Reviewer V285:
> >
> > I would like to begin by expressing my sincere gratitude for the time and effort you have dedicated to reviewing our work. Your insights and feedback are invaluable to us. We hope our responses addressed your questions. If you have any further concerns, we are appreciated and more than willing to provide any further information or clarification.
> >
> > If our responses have addressed your concerns, we would appreciate it a lot if you could improve the score.
> >
> > Thank you once again for your valuable contribution to improving our work. Looking forward to your feedback.

---

> ### Author Response · Authors · 2023-11-22
> **Looking forward to your feedback**
>
> Dear Reviewer V285:
>
> Thank you for your time and effort in reviewing our work. Your insights are invaluable for contributing to our work. If you have more questions or concerns, we're happy to provide further information.
>
>
> Looking forward to your feedback.

---

### Official Review · Reviewer_mLb8 · 2023-11-05

**Soundness:** 3 good
**Presentation:** 3 good
**Contribution:** 2 fair
**Rating:** 6
**Confidence:** 2

**Summary:**

The paper studies the training of neural networks for multi-label learning. The analysis aims to show a neural collapse-type phenomenon when minimizing a pick-all loss function to address the multi-label learning task. By treating the features of every sample as a free optimization variable, the paper formulates and analyzes the optimization problem in equation (4), for which they characterize the global optima (Theorem 1) and show that all local optimal will be globally optima (Theorem 2). Several numerical results are discussed in section 4 to measure the M-lab ETF in training the neural network on multi-label learning tasks.

**Strengths:**

1- The paper is well-written and easy to follow. The authors present their results clearly, and the writing is overall in great shape.

2- The paper focuses on the interesting subject of neural net training dynamics in multi-label learning tasks.

**Weaknesses:**

1- While I understand that analyzing the problem in (3) could be challenging in the general case, I think the simplification of treating every feature vector $h_i = \phi_\theta(x_i)$ as a free optimization variable is restrictive. Could the analysis extend to a more general choice of $\phi_\theta$, e,g, a one-layer overparameterized neural network with some activation function? The authors may still be able to show a weaker result on local optima or stationary points of the objective.  If not, the paper should discuss why the analysis will be challenging for a parameterized neural net function $\phi_\theta$ and some more evidence of why the assumption of treating the features as free variables would make sense for deep neural nets.

2- The theoretical results only analyze the critical points of the loss function, and the paper has no statement on how a gradient-based optimization method will perform in solving (3) or (4). Stating a corollary or theorem on the convergence behavior of a first-order algorithm for optimizing (4) wold be a nice addition. I think Theorems 1,2 could connect the first or second-order stationary points of the objective to the global minima, and so the authors should be able to use the results in the optimization literature to state such a convergence guarantee for a gradient-based optimization algorithm.

**Questions:**

Please see my comments for weaknesses.

---

> ### Author Response · Authors · 2023-11-16
> **Rebuttal by Authors**
>
> We thank the reviewer’s appreciation of our work, finding our result well-presented and interesting. The reviewer’s comments are very thoughtful, we carefully address each question in detail in the following.
>
> $\textbf{Q1:}$  “While I understand that analyzing the problem in (3) could be challenging in the general case, I think the simplification of treating every feature vector
> $h_i=ϕ_θ(x_i$) as a free optimization variable is restrictive. The paper should show some more evidence of why the assumption of treating the features as free variables would make sense for deep neural nets.”
>
> $\textbf{A1:}$ The simplification is a common assumption for analyzing deep neural networks, and has recently been well-studied in the literature; see the references [1,2,3,4]. The motivating reasoning is that deep networks are often highly overparameterized with the capacity to learn arbitrary representations [1,5]. As such, the last-layer features can fit any set of points in the feature space.
>
> This assumption has been empirically validated in previous works for multi-class classification (e.g., Figure 4 in [1] and Figure 2 in [3]). Those experiments show that, even with replacing all the correct labels for each training sample with a random counterpart, if the network is sufficiently overparameterized, the features of the last layer will still exhibit severe neural collapse regardless of the input. For multi-label learning, this phenomenon happens in a more general way: (i) the multiplicity-1 features are maximally separated, and (ii) higher multiplicity features are scaled average of their component features, followed by a max-margin linear classifier.
>
> $\textbf{Q2:}$ “Could the analysis extend to a more general choice of $ϕ_θ$, e,g, a one-layer overparameterized neural network with some activation function? The authors may still be able to show a weaker result on local optima or stationary points of the objective. If not, the paper should discuss why the analysis will be challenging for a parameterized neural net function $ϕ_θ$”
>
> $\textbf{A2:}$ We extend our appreciation to the reviewer for this valuable suggestion. In the context of multi-class classification, there have been recent studies that delve into deeper networks, such as [4] and [6]. These investigations typically assume the input of the network are unconstrained features and analyze the behavior of multi-layer linear networks when applied to such unconstrained features. However, it's important to note that these findings are contingent upon the assumption of unconstrained input features, which overlooks the inherent structures within real-world data inputs. When we consider scenarios where input features are not unconstrained, recent research has demonstrated a progressive feature compression phenomenon across network layers, as evidenced by [7, 8]. As such, neural collapse tends to manifest at the final network layer. Consequently, a single-layer nonlinear network may lack the expressive capacity required to precisely induce neural collapse.
>
> In light of this, we posit that a promising starting point for exploration is to study deep linear networks or nonlinear homogeneous networks which are capable of compressing features across multiple layers. We believe that the insights gained from these studies can be extended from the multi-class setting to the multi-label setting we have explored in this work, thereby warranting further investigation.

---

> ### Author Response · Authors · 2023-11-16
> **Rebuttal by Authors**
>
> $\textbf{Q3:}$ “The theoretical results only analyze the critical points of the loss function, and the paper has no statement on how a gradient-based optimization method will perform in solving (3) or (4). Stating a corollary of the theorem on the convergence behavior of a first-order algorithm for optimizing (4) would be a nice addition. I think Theorems 1,2 could connect the first or second-order stationary points of the objective to the global minima, and so the authors should be able to use the results in the optimization literature to state such a convergence guarantee for a gradient-based optimization algorithm.”
>
> $\textbf{A3:}$ We express our gratitude to the reviewer for the valuable suggestion. In our paper, we establish the theoretical properties of all critical points, demonstrating that the function is a $\textbf{strict saddle function}$ [9] in the context of multi-label learning with respect to (W, H). It's worth noting that for strict saddle functions like ours, there exists a substantial body of prior research in the literature that provides rigorous algorithmic convergence to global minimizers. In our case, this equates to achieving a global multi-label neural collapse solution. These established methods include both first-order gradient descent techniques [9, 10, 11] and second-order trust-region methods [12], all of which ensure efficient algorithmic convergence. We intend to incorporate this discussion into our paper.
>
>
> [1] Zhu, Zhihui, et al. "A geometric analysis of neural collapse with unconstrained features." Advances in Neural Information Processing Systems 34 (2021): 29820-29834.
>
> [2] Fang, Cong, et al. "Exploring deep neural networks via layer-peeled model: Minority collapse in imbalanced training." Proceedings of the National Academy of Sciences 118.43 (2021): e2103091118.
>
> [3] Graf, Florian, et al. "Dissecting supervised contrastive learning." International Conference on Machine Learning. PMLR, 2021.
>
> [4] Tirer, Tom, and Joan Bruna. "Extended unconstrained features model for exploring deep neural collapse." International Conference on Machine Learning. PMLR, 2022.
>
> [5] Cybenko, George. "Approximation by superpositions of a sigmoidal function." Mathematics of control, signals and systems 2.4 (1989): 303-314.
>
> [6] Hien Dang, Tho Tran, Stanley Osher, Hung Tran-The, Nhat Ho, Tan Nguyen, Neural Collapse in Deep Linear Networks: From Balanced to Imbalanced Data, ICML, 2023.
>
> [7] Hangfeng He, Weijie J. Su, A Law of Data Separation in Deep Learning, PNAS, 2023.
>
> [8] Peng Wang, Xiao Li, Can Yaras, Zhihui Zhu, Laura Balzano, Wei Hu, Qing Qu, Understanding Deep Representation Learning via Layerwise Feature Compression and Discrimination, arXiv:2311.02960, 2023.
>
> [9] Ge, Rong, et al. "Escaping from saddle points—online stochastic gradient for tensor decomposition." Conference on learning theory. PMLR, 2015.
>
> [10] Jin, Chi, et al. "How to escape saddle points efficiently." International conference on machine learning. PMLR, 2017.
>
> [11] Jason D. Lee, Ioannis Panageas, Georgios Piliouras, Max Simchowitz, Michael I. Jordan, Benjamin Recht, First-order methods almost always avoid strict saddle points, Mathematical programming, 2019.
>
> [12] Sun, Ju, Qing Qu, and John Wright. "Complete dictionary recovery over the sphere II: Recovery by Riemannian trust-region method." IEEE Transactions on Information Theory 63.2 (2016): 885-914.

---

> ### Author Response · Authors · 2023-11-22
> **Looking forward to your feedback**
>
> Dear Reviewer mLb8:
>
> Thank you for dedicating time to review our work. Your insights are invaluable to improving our work. If you have any additional questions or concerns, we are more than willing to provide further information.
>
> Looking forward to your feedback.

---

### Official Review · Reviewer_VsPM · 2023-11-18

**Soundness:** 2 fair
**Presentation:** 2 fair
**Contribution:** 1 poor
**Rating:** 3
**Confidence:** 5

**Summary:**

The paper explores neural collapse (NC) phenomenon in multi-label (M-lab) learning using deep neural networks.The main content is：
1.The paper provides theoretical analysis to show multi-label NC is the global solution under the unconstrained feature model.
2.The paper introduces the concept of multi-label equiangular tight frames (ETF) to characterize NC geometry.
3.They  empirically demonstrate multi-label NC on practical networks trained on synthetic and real datasets.

**Strengths:**

1.The paper is well-written and easy to follow. The graphics in this article are very intuitive.
2.This article combines NC and M-lab for the first time，providing a new perspective for the study of multi-label.

**Weaknesses:**

The theoretical part of this article is very similar to [1], lacking significant technological innovation and not sufficiently novel.The two papers share a lot of similarities in  technical approach and theoretical analysis frameworks towards extending and understanding the representation geometry of neural networks in multi-label learning tasks via NC.I don't see any important work in your theoretical section that goes beyond what has been done in [1].So, despite the interesting combination of NC and M-lab， I don't consider it as a contributory work.

Ref:
[1] Zhu, Zhihui, et al. "A geometric analysis of neural collapse with unconstrained features." Advances in Neural Information Processing Systems 34 (2021): 29820-29834.

**Questions:**

Please see weekness.

---

> ### Author Response · Authors · 2023-11-20
> **Rebuttal by Authors**
>
> We appreciate that the reviewer finds our work well-written and easy to follow. We also appreciate the author's recognition that our work provides a new perspective in multi-label learning through NC. We address the reviewer’s major concerns in the following:
>
> $\textbf{Q1:}$ “The theoretical part of this article is very similar to [1], lacking significant technological innovation and not sufficiently novel.The two papers share a lot of similarities in technical approach and theoretical analysis frameworks towards extending and understanding the representation geometry of neural networks in multi-label learning tasks via NC”
>
> $\textbf{A1:}$ While drawing inspiration from [1], our primary results and the methodologies employed to derive them deviate significantly from those presented in [1]. As discussed at the bottom of Page 5 in our paper, given only one last layer classifier $\boldsymbol{W}$, the presence of data samples across various multiplicities greater-than-1 features introduce unique challenges in the multi-label learning setting. This is elaborated in our 16-page long detailed proof with  additional lammas (Lemma 4,5,6,7) on new probabilistic and matrix theory in Appendix C. We provided a lemma to lemma comparison to discuss the differences between our result and that of [1] in Appendix A.
>
> $\textbf{Q2:}$  “I don't see any important work in your theoretical section that goes beyond what has been done in [1]. So, despite the interesting combination of NC and M-lab, I don't consider it as a contributory work.”
>
> $\textbf{A2:}$ Beyond the technical discussion in A1, we emphasize that our work's significance goes beyond technical aspects. Our work shows the ubiquity of Neural Collapse (NC)-type phenomenon, in particular in multi-label learning as evidenced through our experimental and theoretical analyses. Importantly, our work improves understanding of multi-label learning through NC and has the potential to lead to better design of strategies for addressing data imbalances in multi-label learning.

---

> ### Author Response · Authors · 2023-11-22
> **Looking forward to your feedback**
>
> Dear Reviewer VsPM:
>
> Thank you for taking the time to review our work. Your insights are invaluable to refining our work. If you have additional questions or concerns, we are happy to provide further information.
>
> Looking forward to your feedback.